# Loss of Ptpn11 (Shp2) drives satellite cells into quiescence

Joscha Griger[1], Robin Schneider[1], Ines Lahmann[1], Verena Schöwel[2], Charles Keller[3], Simone Spuler[2], Marc Nazare[4], Carmen Birchmeier[1]*

[1]Developmental Biology/Signal Transduction Group, Max Delbrück Center for Molecular Medicine (MDC) in the Helmholtz Society, Berlin, Germany; [2]Muscle Research Unit, Experimental and Clinical Research Center, Charité Medical Faculty and Max Delbrück Center for Molecular Medicine Berlin, Berlin, Germany; [3]Children's Cancer Therapy Development Institute, Beaverton, United States; [4]Medicinal Chemistry, Leibniz Institute for Molecular Pharmacology, Berlin, Germany

**Abstract** The equilibrium between proliferation and quiescence of myogenic progenitor and stem cells is tightly regulated to ensure appropriate skeletal muscle growth and repair. The non-receptor tyrosine phosphatase Ptpn11 (Shp2) is an important transducer of growth factor and cytokine signals. Here we combined complex genetic analyses, biochemical studies and pharmacological interference to demonstrate a central role of Ptpn11 in postnatal myogenesis of mice. Loss of Ptpn11 drove muscle stem cells out of the proliferative and into a resting state during muscle growth. This Ptpn11 function was observed in postnatal but not fetal myogenic stem cells. Furthermore, muscle repair was severely perturbed when Ptpn11 was ablated in stem cells due to a deficit in stem cell proliferation and survival. Our data demonstrate a molecular difference in the control of cell cycle withdrawal in fetal and postnatal myogenic stem cells, and assign to Ptpn11 signaling a key function in satellite cell activity.

*For correspondence: cbirch@mdc-berlin.de

Competing interests: The authors declare that no competing interests exist.

## Introduction

Vertebrate skeletal muscle arises during embryonic, fetal, and adult stages. During embryonic development, a pool of Pax7+ progenitor cells is established that provides a cellular source for myogenesis during subsequent life (*Gros et al., 2005*; *Kassar-Duchossoy et al., 2005*; *Relaix et al., 2005*). The Pax7+ cells give rise to primary fibers in embryos, permit growth of muscle mass during fetal and postnatal muscle development as well as repair of injured muscle in the adult (*Sambasivan et al., 2011*; *Lepper et al., 2011*; *McCarthy et al., 2011*; *Murphy et al., 2011*). Previous work identified distinguishing features of Pax7+ cells at different developmental stages based on characteristics like cell size, shape, response to extrinsic signals (*Cossu et al., 1988*; *Cusella-De Angelis et al., 1994*; *Biressi et al., 2007*), molecular signature (*Biressi et al., 2007*; *Pallafacchina et al., 2010*), dependence on $\beta$-catenin (*Hutcheson et al., 2009*), Nuclear Factor I/X (Nfix) (*Messina et al., 2010*), transcriptional co-activator Hmga2 (*Li et al., 2012*), and position in the satellite cell niche (*Bröhl et al., 2012*). In accordance with the original definition, we refer here to cells wedged between the myofiber membrane and the extracellular matrix as satellite cells and identify them by their anatomical position and Pax7 expression (*Mauro, 1961*; *Seale et al., 2000*). Embryonic, fetal and postnatal Pax7+ cells proliferate, but most reach quiescence when muscle fibers cease to grow by accretion of nuclei (around P21 in mice; *White et al., 2010*). However, even in the muscle of adult sedentary mice, a small proportion of satellite cells remains in the cell cycle

and continues to contribute to fibers, although to a low extent that is variable between muscles and animals (*Chakkalakal et al., 2012*; *Keefe et al., 2015*).

Reversible quiescence is believed to be a fundamental characteristic of adult satellite cells. Quiescence of adult stem cells allows their long-term survival and protects them from proliferative damage caused by mutations due to erroneous DNA replication, and from cellular senescence. Transcriptional profiling has shown that quiescent stem cells express low levels of genes involved in DNA replication, cell cycle progression, and mitochondrial metabolism (*Cheung and Rando, 2013*). The tumor suppressor Rb has an important role in regulating quiescence and its ablation results in a vast increase of muscle stem cells and an accelerated cell cycle re-entry (*Hosoyama et al., 2011*). Maintenance of satellite cell quiescence also depends on signaling and is thus actively maintained. Genetic ablation of *Rbpj* encoding the transcription factor mediating canonical Notch signals results in a depletion of the quiescent satellite cell pool due to spontaneous activation and differentiation (*Bjornson et al., 2012*; *Mourikis et al., 2012*). In addition, ablation of *Sprouty 1*, an inhibitor of FGF signaling, results in a mild elevation of the proportion of satellite cells that remain in the cell cycle (*Chakkalakal et al., 2012*). Furthermore, control of protein translation and phosphorylation of translation initiation factor eIF2α is essential for quiescence (*Zismanov et al., 2016*; *Crist et al., 2012*). The mechanisms that allow satellite cells to enter quiescence in postnatal development are less well characterized but known to depend on Hey1 and Heyl, which might act downstream of Notch signals, as well as on the cell cycle inhibitor p27 (*Fukada et al., 2011*; *Chakkalakal et al., 2014*).

Ptpn11 (Shp2) is a tyrosine phosphatase that is an essential and positive mediator of signals provided by many receptors (*Grossmann et al., 2010*; *Dance et al., 2008*; *Neel et al., 2003*). Functions of Ptpn11 downstream of tyrosine kinase receptors are highly conserved in evolution and can be observed in invertebrates and vertebrates (*Perkins et al., 1992*; *Feng et al., 1993*). Ptpn11 frequently acts through regulation of Ras/Mapk/Erk1/2. In many cases, Ptpn11 is required to sustain but not to initiate Erk1/2 activity, indicating that it mainly modulates feed-back regulation of the Ras/Mapk/Erk1/2 branch of the signaling cascade (*Grossmann et al., 2010*; *Dance et al., 2008*; *Neel et al., 2003*). In addition, Ptpn11 was implicated in PI3K, Jak/Stat, Mapk/p38, NF-κB and NFAT signaling in a cell type- and receptor-specific manner (reviewed in *Grossmann et al., 2010*). In the context of myogenesis, Ptpn11 acts during migration of embryonic progenitor cells from somites into the limbs where it mediates signals provided by Met and its adaptor molecule Gab1 (*Schaeper et al., 2007*). In addition, Ptpn11 was assigned a role in hypertrophy as a regulator of NFAT activity (*Fornaro et al., 2006*).

To gain insight into signaling mechanisms that control quiescence, activation and proliferation of myogenic cells, we ablated *Ptpn11* in myogenic progenitor and satellite cells in late embryonic development and the adult. We found that Ptpn11 is dispensable for proliferation in fetal, but not postnatal myogenesis. In particular, satellite cells in the early postnatal period or after regeneration rapidly proliferate. However, when Ptpn11 is absent or inhibited, satellite cells withdraw from the cell cycle and enter a resting state. In culture, satellite cells are not correctly activated when *Ptpn11* is mutated. In particular, *Ptpn11* mutant cells in such cultures upregulate MyoD and therefore appear to enter an activated state, but their proliferation is impaired and they quickly withdraw from the cell cycle. Finally, in the acutely injured muscle, loss of Ptpn11 also impairs survival of satellite cells. Our data demonstrate that ablation or inhibition of Ptpn11 promotes satellite cell quiescence and provides evidence for an unexpected molecular difference in regulation of proliferation in fetal and postnatal myogenic progenitors cells.

## Results

### Ptpn11 controls myogenic stem cell proliferation in postnatal mice

We used a *Pax7*^*Cre*^ allele to introduce conditional *Ptpn11* mutations in the myogenic lineage (*Figure 1—figure supplement 1a*; cf. *Keller et al., 2004*; *Grossmann et al., 2009*). Limb myogenic progenitor cells were isolated by FACS from fetal and postnatal mice carrying hetero- and homozygous conditional mutations of *Ptpn11* (*Pax7*^*Cre*^;*Ptpn11*^*flox/+*^ and *Pax7*^*Cre*^;*Ptpn11*^*flox/flox*^, called control and coPtpn11 mutants, respectively. In some experiments, an additional indicator allele *Rosa*^*eYFP*^ was used; *Figure 1—figure supplement 1b–e*). Analysis of Ptpn11 protein by western blotting showed that it was present in stem cells isolated from fetal and postnatal muscle of control mice and strongly

reduced in cells from coPtpn11 mutants (*Figure 1a*). Thus, *Pax7^Cre* efficiently recombined the *Ptpn11* locus.

We analyzed the consequences of the coPtpn11 mutation, and observed that mutant mice displayed little change in muscle size at late fetal stages (E18) but a severely compromised postnatal muscle growth (*Figure 1b*). Compared to control mice, the numbers of nuclei per fiber were reduced at P7 and P14 but little affected at P0 (*Figure 1c*). In accordance, the diameter of muscle fibers was unchanged at P0 (average fiber diameter 7.9 ± 0.5 μm and 7.9 ± 0.2 μm in control and coPtpn11 mutants, respectively) but significantly reduced at P7 (15.5 ± 0.1 μm and 12.5 ± 0.3 μm in control and coPtpn11 mutants, respectively) and P14 (18.2 ± 0.1 μm and 14.3 ± 0.3 μm in control and coPtpn11 mutants, respectively; see also *Figure 1d*). In addition, the animals developed kyphosis by P14 and therefore subsequent analyses were limited to P7 or earlier developmental stages. To define the cellular basis of the observed deficit in muscle growth, we compared the number of Pax7-positive stem cells in late fetal and postnatal muscle of control and coPtpn11 animals by immunohistology. We observed no significant difference at E16 but at subsequent stages the number of Pax7+ cells declined drastically in coPtpn11 mice, which was confirmed by FACS (*Figure 2a,b*; *Figure 2—figure supplement 1a,b*). We also analyzed animals without a *Pax7^Cre* allele (*Ptpn11^{flox/flox}*) as additional control genotype, and observed no difference in Pax7+ cell numbers between *Ptpn11^{flox/flox}* and *Pax7^Cre; Ptpn11^{flox/+}* mice (*Figure 2b*; *Figure 2—figure supplement 1b*). We conclude that ablation of *Ptpn11* in myogenic progenitor cells has a severe effect on the size of the postnatal stem cell pool.

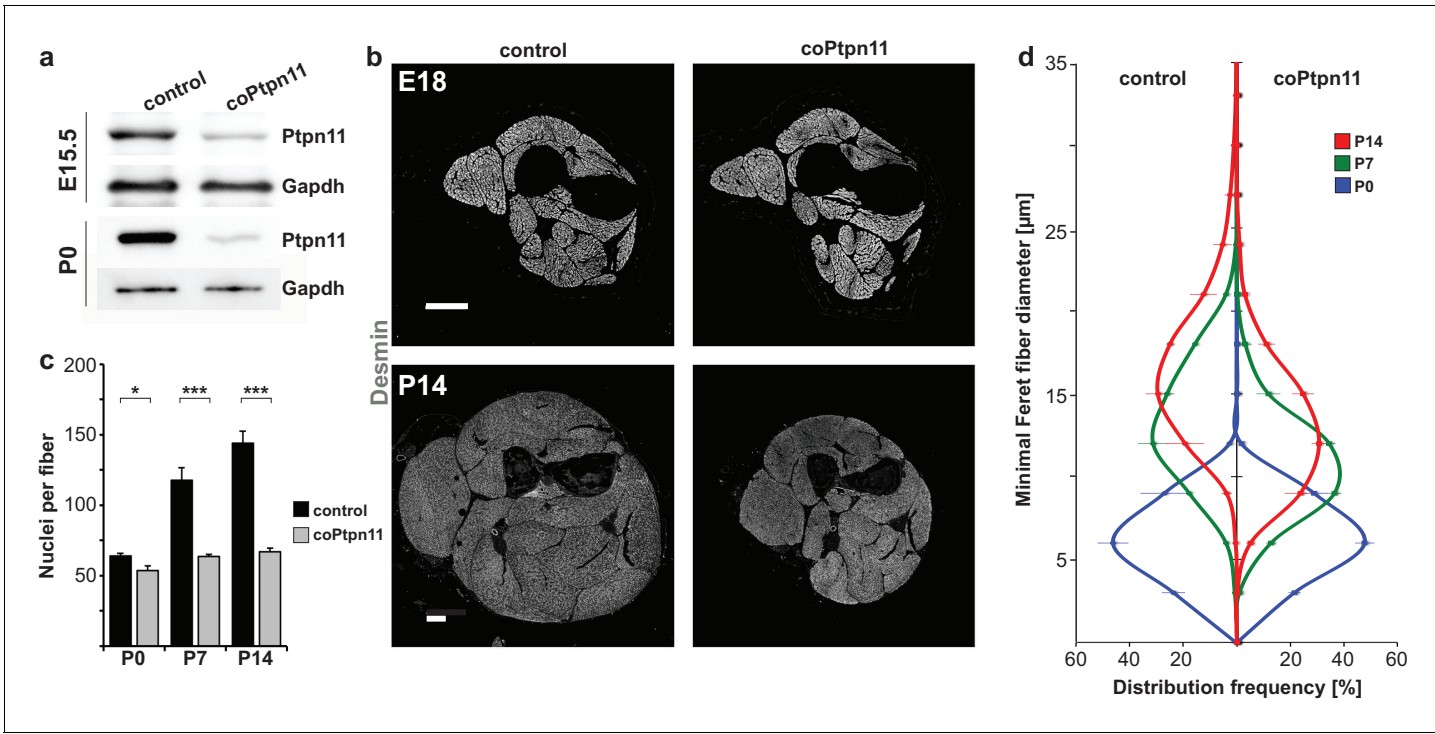

**Figure 1.** Conditional *Ptpn11* mutation leads to a deficit in postnatal muscle growth. (**a**) Western blot analysis of Ptpn11 in YFP-positive cells isolated by FACS from limbs of control and coPtpn11 mutant mice that carry the *Rosa^{eYFP}* allele; YFP-positive cells from E15.5 and P0 animals were analyzed. (**b**) Histological analysis of the lower forelimb of control and coPtpn11 mutant mice at E18 and P14 using anti-desmin antibodies. (**c**) Quantification of nuclei per *extensor carpi radialis longus* muscle fiber at P0, P7 and P14. (**d**) Minimal Feret fiber diameter distribution of *extensor carpi radialis longus* myofibers at P0, P7, P14. *p<0.05, **p<0.01, ***p<0.001. Error bars show S.E.M. Scale bar: 250 μm.

The following figure supplement is available for figure 1:

**Figure supplement 1.** Genetic strategy used to mutate *Ptpn11* and isolation of myogenic stem cells.

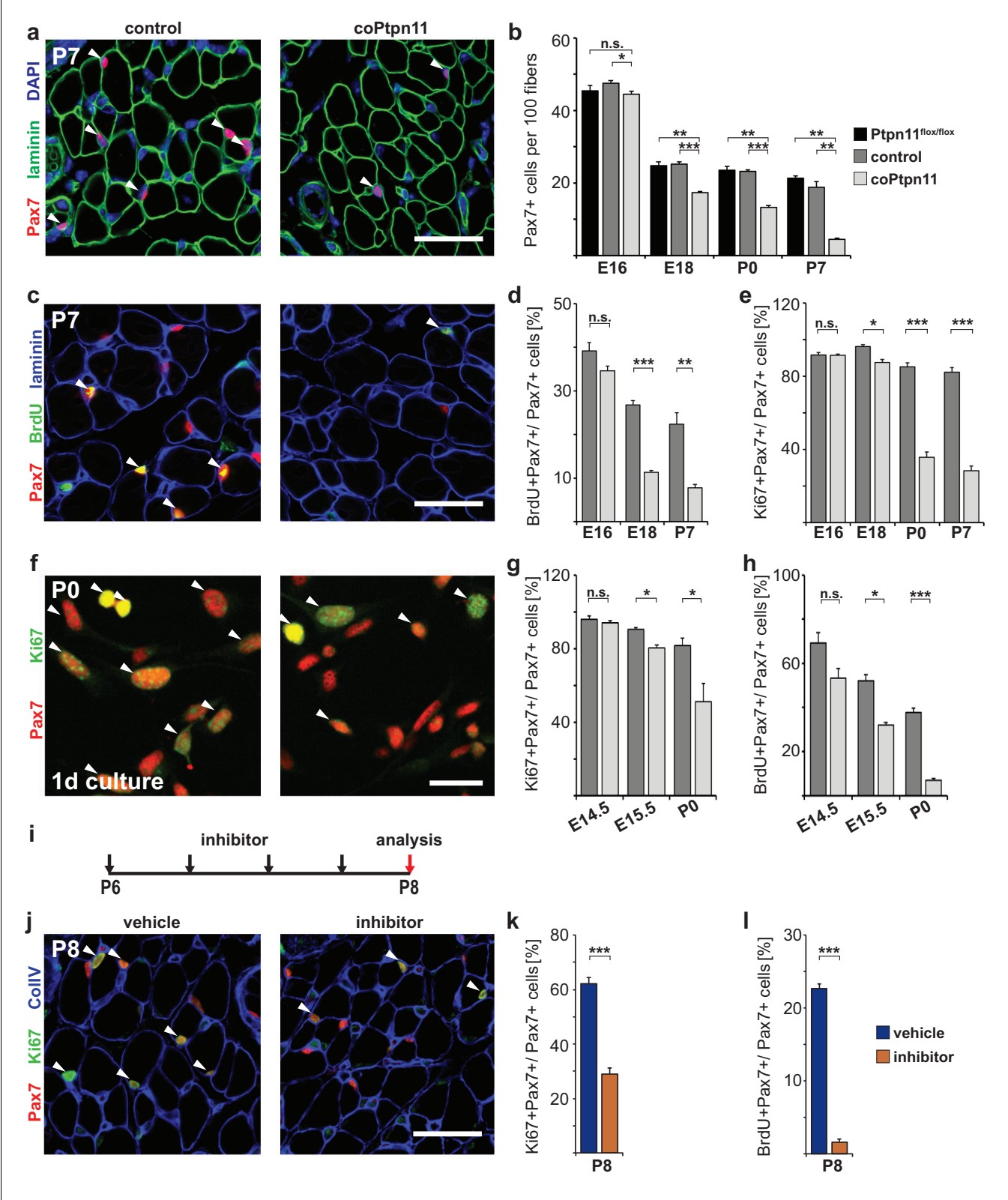

**Figure 2.** Ptpn11 is essential for proliferation of muscle stem cells in neonatal mice. (a) Immunohistological analysis of Pax7 (red) and laminin (green) in muscle of control and coPtpn11 mutant mice; DAPI (blue) was used as a counterstain. Arrowheads point to Pax7+ cells. (b) Quantification of Pax7+ cells per 100 fibers. (c) Immunohistological analysis of Pax7 (red), laminin (blue) and BrdU (green) in forelimb muscles of control and coPtpn11 mutants at P7. Arrowheads point to Pax7+BrdU+ cells. (d,e) Quantification of Pax7+ cells that incorporated BrdU (d) or co-expressed Ki67 (e) at E16, E18, P0 and P7 in

*Figure 2 continued on next page*

*Figure 2 continued*

control and coPtpn11 mutants; the BrdU pulse was given for 2 hr. (f) Pax7 (red) and Ki67 (green) immunostaining of cultured myogenic progenitor cells isolated from control and coPtpn11 mutant mice that carry the *Rosa*^*eYFP*^ allele; YFP-positive cells were isolated by FACS and cultured for 18–20 hr. Arrowheads show Pax7+Ki67+ cells. (g,h) Quantification of cultured Pax7+ cells that incorporate BrdU (g) or co-express Ki67 (h) at E14.5, E15.5 and P0; the BrdU pulse was given for 1 hr. (i) Scheme for GS493 injections during postnatal growth. (j) Immunohistological analysis of Pax7 (red), Ki67 (green) and collagen IV (ColIV, blue) in forelimb muscles (P8) of animals treated with GS493 or vehicle from P6 to P8. Arrowheads point to Pax7+Ki67+ cells. (k, l) Quantification of Pax7+ cells that co-express Ki67 (k) or incorporated BrdU (l) in muscle of GS493- or vehicle-treated animals. n.s. = not significant, *p<0.05, **p<0.01, ***p<0.001. Error bars show S.E.M. Scale bars 25 μm in a, c, j, 15 μm in f.

The following figure supplements are available for figure 2:

**Figure supplement 1.** *Ptpn11* mutation leads to decreased numbers of myogenic progenitor cells but does not affect apoptosis.

**Figure supplement 2.** *Ptpn11* mutation impairs proliferation of satellite cells.

To assess the mechanism responsible for the change in Pax7+ cell numbers, we analyzed proliferation. BrdU incorporation and Ki67 expression in Pax7+ cells were similar in control and coPtpn11 mice at E16.5, but at subsequent stages the numbers were severely reduced in mutants (*Figure 2c–e*). Ki67 is expressed in proliferating cells in all stages of the cell cycle but not found in resting cells (*Scholzen and Gerdes, 2000*). The downregulation of Ki67 observed here indicates that the muscle stem cells withdraw from the cell cycle prematurely. No change in apoptosis rates was observed by TUNEL staining (*Figure 2—figure supplement 1c–e*).

*Pax7*^*Cre*^ introduces mutations into myogenic progenitors and their descendants. Thus, stem cells as well as fiber nuclei lack Ptpn11 in the mutant mice. To assess whether proliferation was impaired in a cell-autonomous manner, myogenic progenitors from coPtpn11 and control mice were isolated, cultured and proliferation was assessed. BrdU incorporation and Ki67 expression were compromised in cultured coPtpn11 mutant progenitors isolated from P0 but not from E14.5 mice, and an intermediary effect was observed in E15.5 progenitors (*Figure 2f–h*). We conclude that loss of Ptpn11 impairs proliferation in a cell-autonomous manner. This Ptpn11 function is observed in stem cells of the postnatal but not fetal muscle.

We also tested the effect of pharmacological inhibition of Ptpn11 in the early postnatal muscle (GS493; cf. *Grosskopf et al., 2015*). When mice were treated for two days with the inhibitor, the number of Pax7+ cells that co-expressed Ki67 or incorporated BrdU was severely reduced (*Figure 2i–l*). Similar to the conditional mutant mice, no change in apoptosis rates was observed by TUNEL staining (*Figure 2—figure supplement 1f,g*). We conclude that in the absence of Ptpn11, early postnatal stem cells prematurely withdraw from the cell cycle.

We analyzed differentiation in stem cells in which Ptpn11 was ablated or inhibited. Similar proportions of freshly FACS-isolated stem cells were Pax7- and MyoG-positive, regardless whether cells from control and coPtpn11 mutants or from control and GS493-treated mice were compared, indicating that their differentiation propensity was unaffected (*Figure 3a–c*). Furthermore, FACS-isolated coPtpn11 mutant cells displayed no obvious fusion deficit when cultured in differentiation medium (*Figure 3d*). Thus, postnatal stem cells that lack Ptpn11 activity are impaired in proliferation but neither their differentiation capacity nor the probability of their differentiation are altered. Together the data indicate that the pool of Pax7+ cells is not replenished by proliferation. Furthermore, MyoG expression indicates that a similar proportion of control and coPtpn11 myogenic cells differentiate continuously, leading thus to a pronounced decrease in the number of Pax7+ cells over time.

Next we performed gene expression profiling of neonatal stem cells from control and coPtpn11 mice. 752 differentially expressed genes were identified (FC> ± 1.5; q < 0.05; see **Supplementary file 1** for the complete list of deregulated genes). GO term enrichment analysis demonstrated that down-regulated genes encode components of various stages of the cell cycle (e.g. G1/S-specific *Ccne2*; G1/S, S and M-specific *Ranbp1*; S, G2 and M phase specific *Prc1*; G2/M-phase specific *Cdk1*; M-phase-specific *Ccnb1* and *Nusap1*; see also **Supplementary file 1, 2**). We also noted that among the ten most strongly down-regulated genes, four are known as immediate early genes that respond to growth factors (*Egr4, Tnfrsf12a, Egr2, Plk2*; **Table 1**; **Supplementary file 1**; *Meighan-Mantha et al., 1999*; *Simmons et al., 1992*; *Müller et al., 1991*;

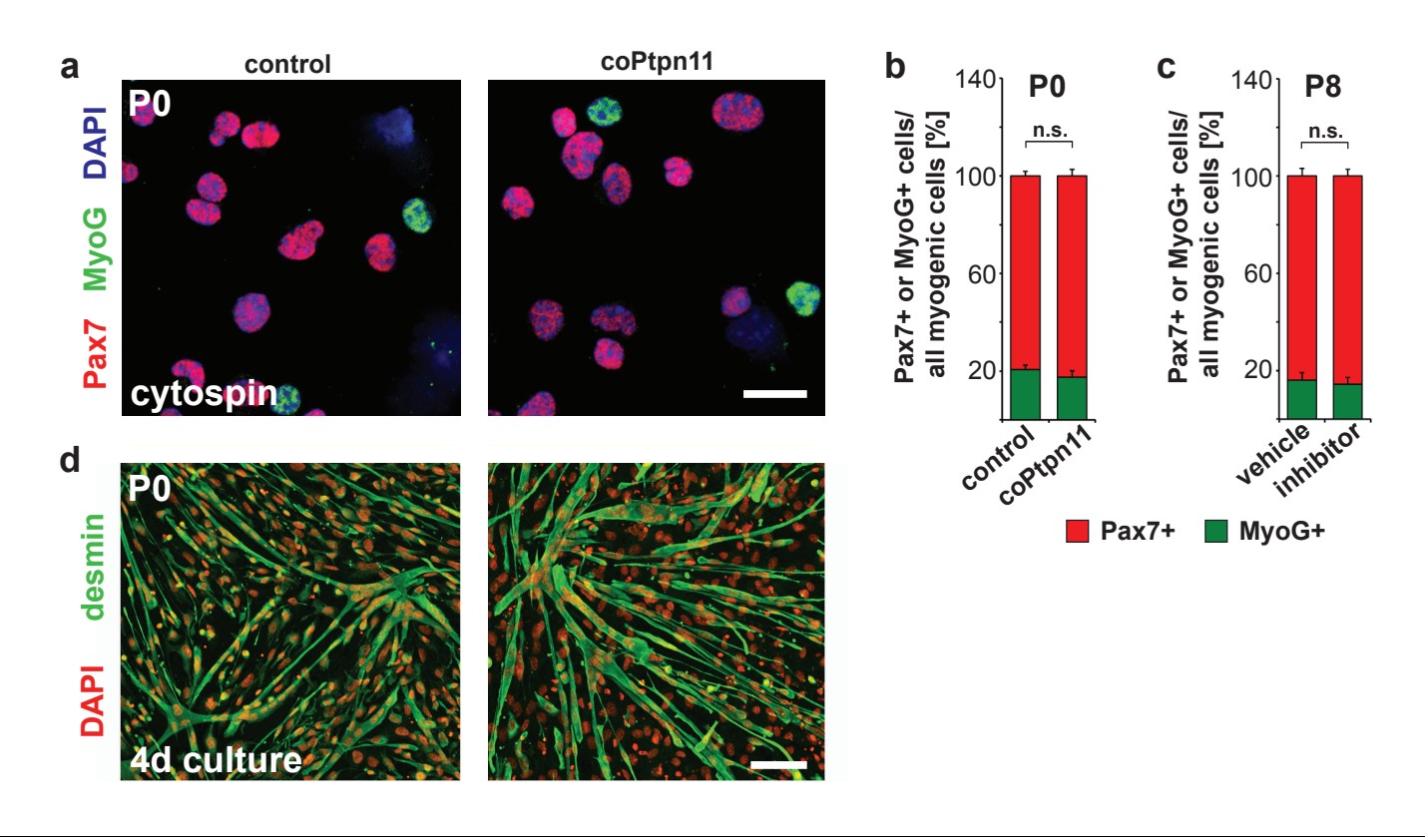

**Figure 3.** Loss of Ptpn11 does not affect myogenic differentiation. (a) Cytospins of freshly isolated muscle stem cells from control and coPtpn11 mice (P0); cells were immunostained for Pax7 (red) and MyoG (green), and nuclei were counterstained with DAPI (blue). (b) Quantification of Pax7+ and MyoG+ cells in cytospins of cells isolated from control and coPtpn11 mice (P0). (c) Quantification Pax7+ and MyoG+ cells in cytospins of freshly isolated stem cells after donors had been treated with GS493 or vehicle. (d) Immunostaining for desmin (green) of cultured myogenic progenitor cells kept in differentiation medium for 4 days. Nuclei were counterstained with DAPI (red). n.s. = not significant. Error bars show S.E.M. Scale bars 10 μm (a) 50 μm (d).

*Gashler and Sukhatme, 1995*). Among the up-regulated genes, many encode extracellular matrix proteins and proteases that remodel extracellular matrix (e.g. *Col1a1/2, Col16a1, Adamts2*; *Table 1*; *Supplementary file 1*). We used PCR to verify changes in the expression of markers of the cell cycle (*Figure 2—figure supplement 2a*) that are in accordance with the reduced proliferation and premature cell cycle exit of the myogenic stem cells. Moreover, this was supported by a cell cycle profile analysis (*Figure 2—figure supplement 2c,d*). In addition, we quantified autophagy- and senescence related transcripts, myogenic transcription factors, as well as markers expressed in endothelia and fat (*Figure 2—figure supplement 2a,b*), which excluded that coPtpn11 mutant cells undergo autophagy, senescence, differentiate prematurely into muscle, or trans-differentiate into endothelia/adipocytes. Lastly, the observed downregulation of immediate early genes indicated that *Ptpn11* mutant cells no longer respond appropriately to growth factor signaling.

## Ptpn11 regulates sustained Erk1/2 activity in myogenic cells and is necessary for correct activation of satellite cells

Ptpn11 effects on signaling were reported to vary in a context-dependent manner (*Grossmann et al., 2010*). We used GS493 mediated inhibition of Ptpn11 in C2C12 cells to systematically investigate the effects of Ptpn11 on the activity of signaling molecules in myogenic cells. GS493 strongly inhibited Erk1/2 phosphorylation (pErk1/2) and thus the activity of the Mapk/Erk1/2 pathway in C2C12 cells, but phosphorylation of Akt and p38 were unchanged (*Figure 4a*). After starvation of C2C12 cells, addition of serum quickly up-regulated pErk1/2 in the presence or absence of

**Table 1.** Gene expression changes in *Ptpn11* mutant satellite cells.

| Cell cycle associated genes | FC | FDR |
|---|---|---|
| Ccnd1 Cyclin D1 | -2.58 | 0.00363 |
| Ccnb1 Cyclin B1 | -2.33 | 0.00144 |
| Prc1 Protein regulator of cytokinesis 1 | -2.32 | 0.01522 |
| Ranbp1 Ran-binding protein 1 | -2.18 | 0.00081 |
| Cdc2a Cyclin dependent kinase 1a | -2.11 | 0.00402 |
| Immediate early genes | | |
| Egr4 Early growth response 4 | -3.00 | 0.00300 |
| Tnfrsf12a FGF-inducible 14/ Tweak-receptor | -2.65 | 0.00144 |
| Egr2 Early growth response 2 | -2.58 | 0.00318 |
| Plk2 Polo-like-kinase 2 | -2.57 | 0.00004 |
| Extracellular matrix/ secreted proteins | | |
| Adamts2 ADAM Metallopeptidase 2 | 3.15 | 0.00144 |
| Thbs2 Thrombospondin 2 | 2.89 | 0.00519 |
| Mfap5 Microfibrillar-associated protein 4 | 2.84 | 0.01179 |
| Col1a1 Collagen 1a1 | 2.75 | 0.00374 |
| Mfap2 Microfibrillar-associated protein 2 | 2.62 | 0.00179 |
| Col1a2 Collagen 1a2 | 2.29 | 0.00082 |
| Col16a1 Collagen 16a1 | 2.17 | 0.00681 |

FC: fold-change; FDR: false discovery rate; corrected p-value (q-value) determined as described (Hochberg and Benjamini, 1990).

the inhibitor. However, sustained pErk1/2 levels were severely decreased by GS493 (*Figure 4b*). Downregulation of pErk1/2 levels was also observed in isolated satellite cells from coPtpn11 mutant mice (*Figure 4c*, quantified in d). Thus, Ptpn11 is required in myogenic cells for sustained activation of Mapk/Erk1/2.

To analyze Ptpn11 functions genetically in adult satellite cells, we used an tamoxifen-dependent Cre allele to introduce conditional *Ptpn11* mutations into 5-week-old mice (called TxcoPtpn11 mutant mice that correspond to *Pax7^CreERT2^;Ptpn11^flox/flox^* mice (*Lepper et al., 2009*); tamoxifen treated *Pax7^CreERT2^;Ptpn11^flox/+^* mice served as controls and are called Txcontrol; *Figure 4—figure supplement 1a*). qPCR analysis of *Ptpn11* transcripts isolated from satellite cells of TxcoPtpn11 mice demonstrated efficient recombination after tamoxifen administration (*Figure 4—figure supplement 1b,c*). Floating fiber cultures serve as a model to monitor activation, proliferation, self-renewal and differentiation of satellite cells (*Zammit et al., 2004*). We isolated fibers from TxcoPtpn11 and Txcontrol mice 10 days after tamoxifen treatment, and analyzed phosphorylation of Erk1/2 and S6 protein as well as appearance of MyoD protein; appearance of pS6 and MyoD were described as early events during satellite cell activation (*Rodgers et al., 2014*). pS6 and MyoD were present in satellite cells on fibers isolated from control and TxcoPtpn11 mice after 24 hrs of culture, whereas pErk1/2 staining was strongly reduced (*Figure 4e–g*; *Figure 4—figure supplement 1d*). After 48 hrs, a substantial proportion of satellite cells from control and TxcoPtpn11 mice had entered the cell cycle and expressed Ki67 (*Figure 5a,b*). After 72 hrs, the majority of control satellite cells had undergone multiple cell divisions and formed small colonies, but satellite cells from TxcoPtpn11 mice divided rarely, most were BrdU-negative and had downregulated Ki67, indicating that they had withdrawn from the cell cycle (*Figure 5c,d,f,g*; *Figure 5—figure supplement 1a,b*). We next asked whether sustained Mapk/Erk1/2 signaling rescues proliferation when *Ptpn11* is ablated. For this, we used an allele (*Gt(Rosa)26Sor^tm8(Map2k1*,EGFP)Rsky^* hereafter called *Map2k1^DD^*) encoding a constitutively active Map2k1 variant, expressed from the *ROSA26* locus; *Map2k1^DD^* coding sequences were preceded by a floxed translational stop cassette (*Figure 4—figure supplement 1a*); the animals used for isolation of fibers carried in addition the *Pax7^CreERT2^* and *Ptpn11^flox^* alleles. Thus, tamoxifen

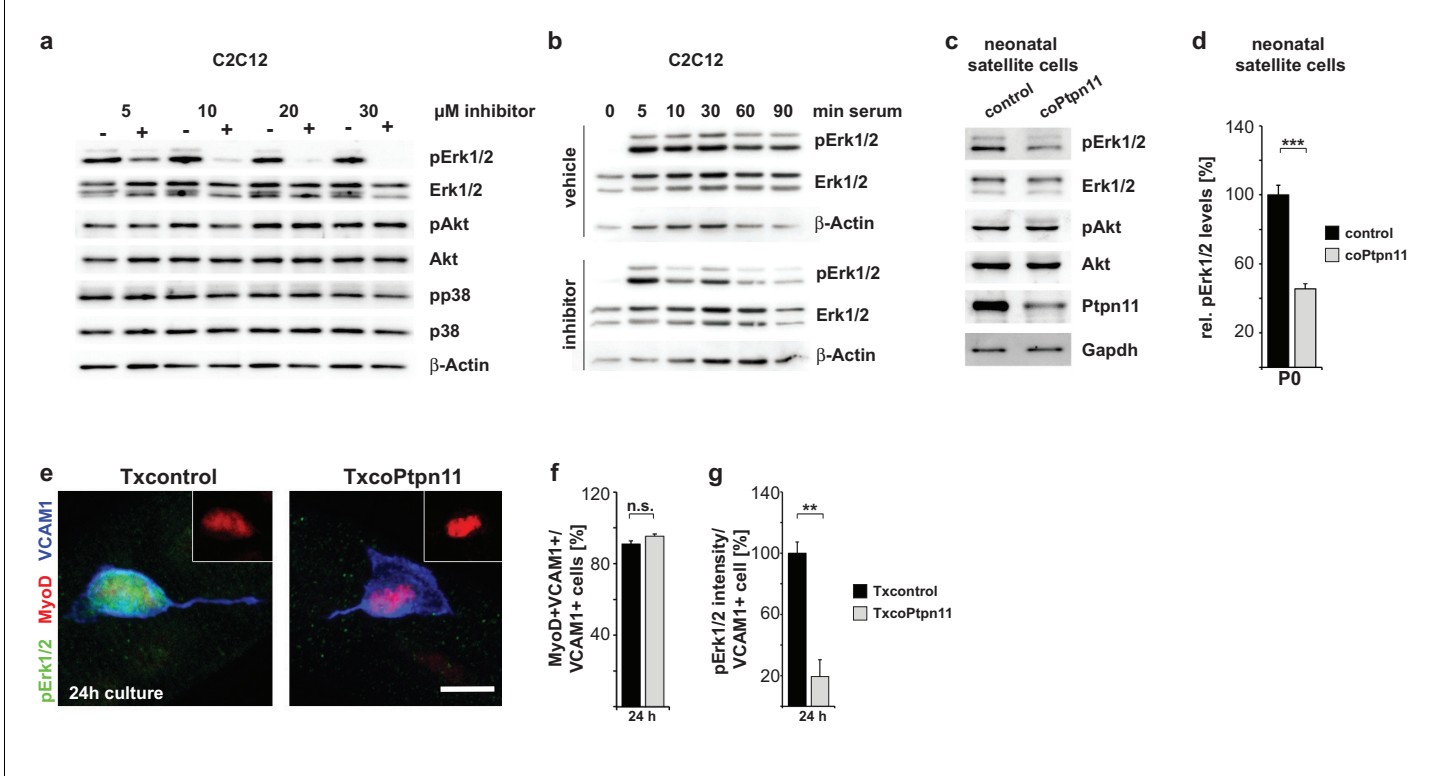

**Figure 4.** Ptpn11 controls Mapk/Erk1/2 activation in myogenic cells. (**a**) Western blot analysis of phosphorylation of Mapk/Erk1/2, PI3K/Akt and Mapk/p38 in C2C12 cells cultured in the presence of either vehicle or GS493 (5, 10, 20 and 30 µM). (**b**) Western blot analysis of Mapk/Erk1/2 phosphorylation in C2C12 cells; cells were incubated for 6 hr in serum-free medium, and stimulated with 20% fetal calf serum in the presence/absence of GS493. (**c**) Western blot analysis of Mapk/Erk1/2 and PI3K/Akt in cultured neonatal satellite cells. (**d**) Quantification of Mapk/Erk1/2 phosphorylation. (**e**) Immunostaining of cultured fibers obtained from adult Txcontrol and TxcoPtpn11 mice. Satellite cells associated with the fibers are displayed, and pErk1/2 (green), MyoD (red) and VCAM1 (blue) staining is shown; the inset displays MyoD (red). (**f**) Quantification of VCAM1+ cells co-expressing MyoD. (**g**) Densitometric quantification of pErk1/2 staining intensity in VCAM1+ cells. n.s. = not significant, *p<0.05, **p<0.01, ***p<0.001. Error bars show S.E.M. Scale bar: 10 µm.

The following figure supplement is available for figure 4:

**Figure supplement 1.** Efficient recombination of the *Ptpn11^flox^* allele in adult satellite cells and floating fiber cultures.

---

treatment simultaneously induced cre-dependent ablation of *Ptpn11* and expression of *Map2k1^DD^* in satellite cells. Cultured fibers from such animals demonstrated that Map2k1^DD^ expression rescued the deficits in colony size, BrdU incorporation and Ki67 expression in satellite cells that lack *Ptpn11* (*Figure 5e–g*). We conclude from this experiment that Shp2 is a crucial regulator of Mapk/Erk1/2 activity in satellite cells. We also noted that the proportion of MyoG expressing cells was increased when *Ptpn11* was ablated, whereas sustained Mapk/Erk1/2 activity interfered with differentiation (*Figure 5—figure supplement 1a,c,d*). Thus, in myofiber culture not only proliferation but also differentiation was affected by loss of *Ptpn11*.

## Ptpn11 controls proliferation of adult satellite cells and muscle repair

Next we analyzed the consequences of the adult *Ptpn11* mutation in vivo. A severe decline in satellite cell numbers was observed in TxcoPtpn11 mutants 90 days after introduction of the mutation, but numbers were not significantly changed 10 days after recombination (*Figure 6a–c*; *Figure 6—figure supplement 1a,b*). Thus, the loss of cells observed was slow and corresponded to a disappearance of 0.04 satellite cells/100 fibers/day when TxcoPtpn11 and Txcontrol were compared at P90. We did not observe elevated levels of apoptosis using TUNEL or Caspase activity (*Figure 6—figure supplement 1c–f*). Furthermore, qPCR analyses showed that MyoD, MyoG, markers for fat

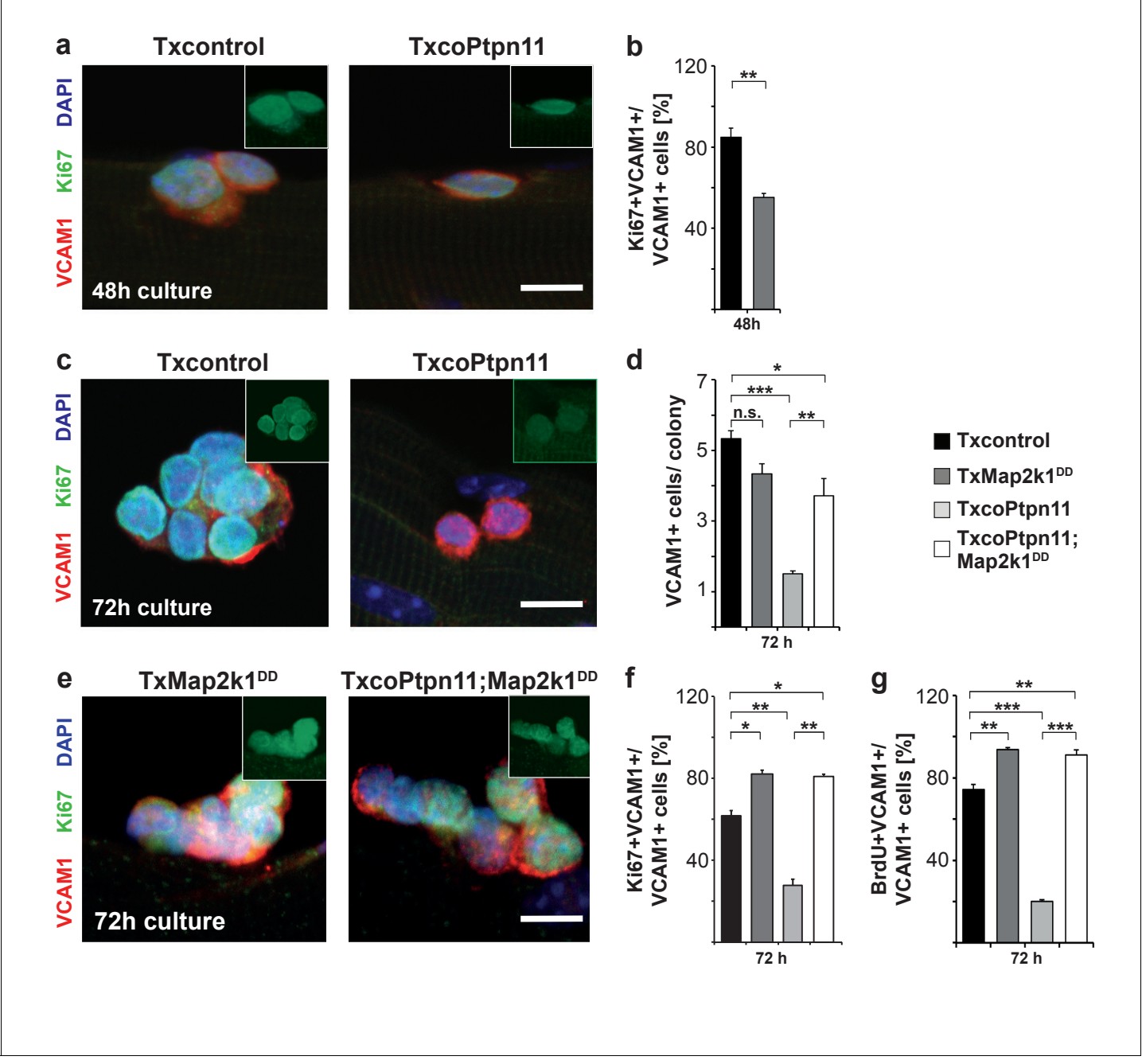

**Figure 5.** Activation of Mapk/Erk1/2 signaling rescues proliferation of *Ptpn11*-mutant satellite cells. Fibers from adult Txcontrol and TxcoPtpn11 mice were cultured for the indicated times, and analyzed for VCAM1 (red) and Ki67 (green) protein; the inset depicts Ki67 staining (green). Nuclei were counterstained with DAPI (blue). (**b**) Quantification of VCAM1+ cells co-expressing Ki67. (**c,e**) Staining of satellite cells for VCAM1 (red), Ki67 (green) and DAPI (blue); fibers were isolated from (**c**) Txcontrol and TxcoPtpn11 mutant mice and (**e**) from TxMap2k1[DD] and TxcoPtpn11;Map2k1[DD] mice. The insets depict Ki67 staining (green). (**d**) Quantification of VCAM1+ cells/colony. (**f,g**) Quantification of VCAM1+ cell co-expressing Ki67 (**f**) or incorporating BrdU (**g**). n.s. = not significant, *p<0.05, **p<0.01, ***p<0.001. Error bars show S.E.M. Scale bar: 10 μm.

The following figure supplement is available for figure 5:

**Figure supplement 1.** Ptpn11/Mapk/Erk signaling affects satellite cell differentiation in floating single myofiber cultures.

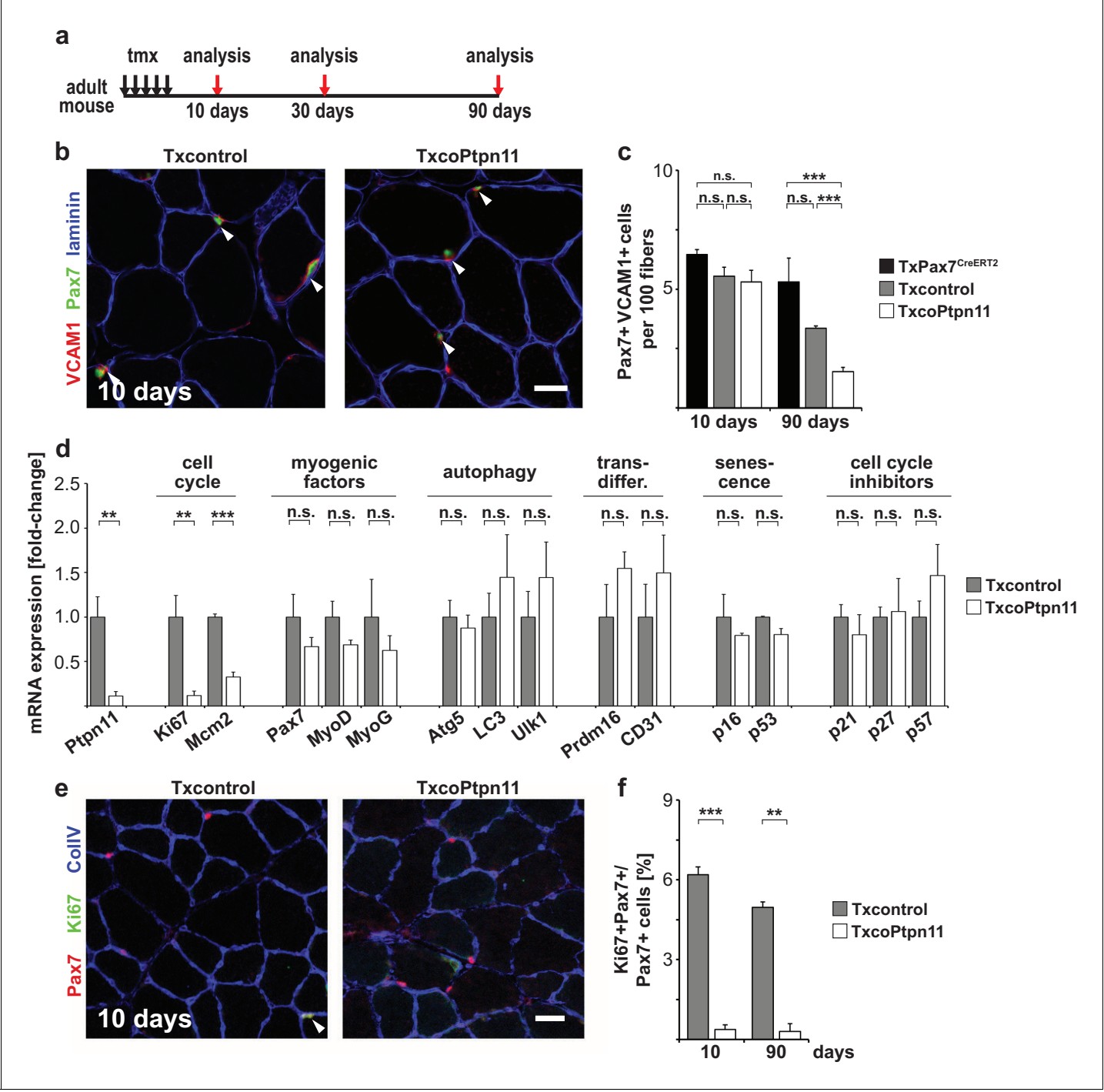

**Figure 6.** Ptpn11 is essential for long-term maintenance of satellite cells in sedentary adult muscle. (**a**) Outline of the experiment. (**b**) Immunostaining for VCAM1 (red), Pax7 (green), and laminin (blue) of *tibialis anterior* muscle from Txcontrol and TxcoPtpn11 mice 10 days after tamoxifen; arrowheads point to Pax7+VCAM1+ cells. (**c**) Quantification of Pax7+VCAM1+ cells in control and TxcoPtpn11 animals 10 and 90 days after tamoxifen. (**d**) Quantification of mRNA transcripts expressed in satellite cells isolated by FACS from control (Txcontrol) and mutant (TxcoPtpn11) mice 30 days after tamoxifen. (**e**) Immunostaining for Pax7 (red), Ki67 (green) and collagen (ColIV, blue) of muscle from Txcontrol and TxcoPtpn11 mice 10 days after tamoxifen. Arrowheads point to Pax7+Ki67+ cells. (**f**) Quantification of Pax7-positive cells co-expressing Ki67 10 and 90 days after tamoxifen administration. n.s. = not significant, *p<0.05, **p<0.01, ***p<0.001. Error bars show S.E.M. Scale bars: 25 μm.

The following figure supplement is available for figure 6:

**Figure supplement 1.** Ptpn11 is essential for long-term maintenance of satellite cells and muscle regeneration.

and endothelial cells, as well as genes related to autophagy or senescence were unchanged in the TxcoPtpn11 mutant satellite cells (*Figure 6d*). However, a significant downregulation of cell cycle markers (Ki67, Mcm2) was observed. Our data indicate that Ptpn11 is essential for long-term maintenance of satellite cells in the adult.

We verified downregulation of Ki67 by immunohistology. The majority of Pax7+ cells in the adult animal are quiescent, but a subpopulation was reported to be in the cell cycle (*Chakkalakal et al., 2012*). In accordance, we observed that Ki67 was expressed in a sub-population of Pax7+ cells in muscle of control sedentary mice, and noted a significantly lowered number of Ki67+Pax7+ cells in TxcoPtpn11 mutants (*Figure 6e,f*). We conclude that ablation of *Ptpn11* drives the subpopulation of satellite cells that retain cell cycle activity into quiescence, and also interferes with long term maintenance of satellite cells. Cell cycle activity might thus play a role in long term maintenance of satellite cells, but the exceedingly slow loss makes it difficult to exclude other mechanisms like cell death or differentiation.

Satellite cells provide the source of stem cells for muscle regeneration, and we tested regeneration in TxcoPtpn11 mutants. A muscle injury was induced 7–10 days after tamoxifen treatment, i.e. at a time point when satellite cell numbers were unaffected by the mutation. Histological analyses indicated that regeneration of the muscle was severely impaired in TxcoPtpn11 mutants, and only few regenerated fibers with small diameter were discernable seven days after injury (*Figure 7a–c*). Instead, fibrotic tissue and fat deposits had accumulated (*Figure 7d*; *Figure 7—figure supplement 1a*). Furthermore, seven days after injury increased numbers of C/EBPα+ adipocytes were observed in the damaged tissue of TxcoPtpn11 mutants (*Figure 7—figure supplement 1b,c*; *Cristancho and Lazar, 2011*). Previous observations by others demonstrated accumulation of fat and fibrotic tissue when muscle regeneration is severely impaired which was assigned to differentiation of fibro-adipogenic progenitors (*Sambasivan et al., 2011*; *Uezumi et al., 2010*; *Lepper et al., 2011*), and we suggest that such a mechanism is responsible for the adipogenesis in our model. In the newly regenerated muscle seven days after injury, satellite cell numbers are increased compared to resting muscle (*Ogawa et al., 2015*). In accordance, we observed many Pax7+Vcam1+ cells in newly regenerated muscle of control mice, but in TxcoPtpn11 mice the Pax7+Vcam1+ satellite cells were rare (*Figure 7e,f*). Thus, satellite cells require Ptpn11 to regenerate muscle tissue and to replenish the satellite cell pool.

To assess the mechanism of the impaired regeneration, we analyzed satellite cells during different stages of the regeneration process. At 2.5 days after injury, the number of satellite cells was strongly reduced in TxcoPtpn11 mutant mice compared to control animals, and seven days after injury the number of satellite cells detected was even lower and had reached levels that were below the numbers present in uninjured muscle (*Figure 7e,f*). As assessed by TUNEL staining, apoptosis levels were increased 2.5 days after injury in TxcoPtpn11 mutants, indicating that during this acute phase satellite cells are lost due to apoptosis (*Figure 7g*). Next we tested whether satellite cells were proliferating in TxcoPtpn11 mutants, and concentrated on an early time point after injury because sufficient numbers of satellite cells for such an analyses were still detectable. This showed that many satellite cells were Ki67+, but very few were positive for the M phase marker phospho-Histone H3 (*Figure 7*h,i). We conclude that in vivo Ptpn11 mutant satellite cell enter an early activation stage, but are unable to proliferate, similarly to the changes observed in floating fiber culture. Furthermore, in a short time window after muscle injury, satellite cells are lost due to apoptosis.

We next asked whether Ptpn11 controls the re-entry of satellite cells into quiescence after regeneration. Because of the very low number of satellite cells in TxcoPtpn11 mutants after muscle regeneration, we used short-term pharmacological inhibition and treated control mice with GS493 seven days after injury. GS493 severely reduced the proportion of Pax7+ cells that remained in the cell cycle and reduced the number of Pax7+ cells present in the newly regenerated muscle (*Figure 8a–e*). In this late phase of the regeneration process, GS493 neither induced increased apoptosis nor did it change the proportion of cells expressing MyoG, or C/EBPα (*Figure 8f,g*; *Figure 8—figure supplement 1a–e*). We conclude that in the late phase of regeneration, satellite cells prematurely re-enter quiescence if Ptpn11 is inhibited.

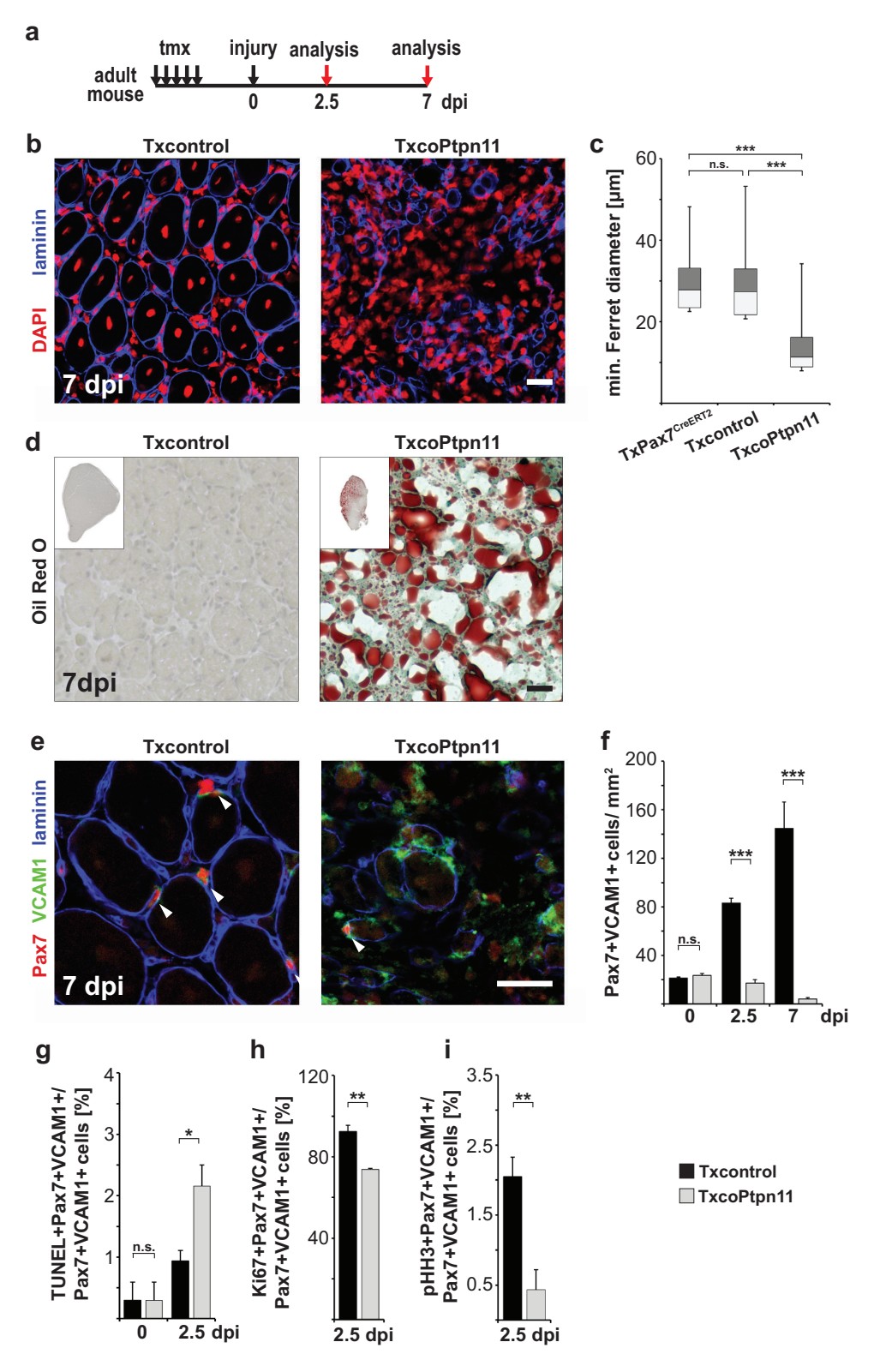

**Figure 7.** Ptpn11 is essential for skeletal muscle repair. (**a**) Outline of the regeneration experiment. (**b**) Immunostaining for laminin (blue) of injured muscle from Txcontrol and TxcoPtpn11 animals seven days after injury (7dpi). Nuclei were counterstained with DAPI (red). (**c**) Box plot showing the quartile distribution and minima/maxima of the diameters of regenerating fibers. (**d**) Oil Red O staining of regenerating muscle seven days after injury. (**e**) Immunohistochemical analysis using antibodies for Pax7 (red), VCAM1 (green) and laminin (blue) of muscle after injury; arrowheads point towards

*Figure 7 continued on next page*

*Figure 7 continued*

Pax7+VCAM1+ cells. (f) Quantification of Pax7+VCAM1+ cells/mm$^2$ before injury, as well as 2.5 and 7 days after injury. (g) Quantification of Pax7+VCAM1+TUNEL+ positive cells before and 2.5 days after muscle injury. (h, i) Quantification of Pax7+VCAM1+ cells co-expressing Ki67 (h) or phospho-Histone H3 (pHH3) (i) 2.5 days after injury. n.s. = not significant, *p<0.05, **p<0.01, ***p<0.001. Error bars show S.E.M. Scale bar: 25 μm.

The following figure supplement is available for figure 7:

**Figure supplement 1.** Mutation of *Ptpn11* in satellite cells leads to a severe muscle regeneration deficit and accumulation of fibrotic and adipose tissue.

## Discussion

The tyrosine phosphatase Ptpn11 is an important transducer of signals provided by growth factors and cytokines (*Grossmann et al., 2010*; *Neel et al., 2003*). We demonstrate here that Ptpn11 plays a central role in myogenesis. Loss of Ptpn11 drives satellite cells in postnatal myogenesis into a resting state. This results in blunted muscle growth and kyphosis. It is interesting to note that patients with hypomorph mutations in *Ptpn11* (Leopard or Noonan syndrome) also display kyphosis, and functional muscle deficits might contribute to the appearance of this phenotype (*NIH-GARD, 2016a*, *2016b*). In addition, we noted that muscle regeneration was impaired, which was assigned to a proliferation deficit and an impaired survival of satellite cells in the initial phase after injury. We combined complex genetic analyses, biochemical studies and pharmacological interference to show that Ptpn11 mainly acts by controlling sustained Mapk/Erk1/2 activity in myogenesis. Our findings identify the tyrosine phosphatase Ptpn11 as a central regulator of satellite cell activity during postnatal muscle growth and regeneration.

### Ptpn11 as regulator of Mapk/Erk1/2 activity in myogenesis

We show that Ptpn11 controls proliferation of postnatal but not fetal myogenic progenitor cells. Thus, despite the fact that postnatal and fetal progenitors express Ptpn11, they differentially depend on Ptpn11 for proliferation. This difference of Ptpn11 functions can be observed not only in vivo but also when the cells were isolated and cultured. Thus, the mechanistic differences in the control of proliferation go beyond a switch in signals that embryonic and fetal progenitors might encounter in vivo. Embryonic, fetal and postnatal myogenic cells were previously shown to differ in cellular properties, gene expression and response to signaling molecules (*Biressi et al., 2007*; *Kassar-Duchossoy et al., 2005*; *Li et al., 2012*). Our data indicate that a major switch exists in the signaling that controls proliferation during myogenesis.

Using pharmacological inhibition, we show that Ptpn11 regulates the activity of Erk1/2 but not other tested signaling molecules in C2C12 cells. In particular, sustained Erk1/2 activity was blunted, but Erk1/2 short-term activation was unchanged. This downregulated Erk1/2 activity was also observed in stem cells that lack *Ptpn11*. Erk1/2 inhibition was previously described to interfere with proliferation in fetal and adult myogenic cells and, conversely, increasing Mapk/Erk1/2 activity by ablation of *Sprouty1/2*, enhances proliferation at both stages (*Michailovici et al., 2014*; *Lagha et al., 2008*). Thus, Mapk/Erk1/2 but not Ptpn11 functions are stage-independent.

We show that the numbers of muscle stem cells decreased in the postnatal but not the fetal muscle when *Ptpn11* is mutated, and we assigned this to a pronounced proliferation deficit. Thus, the pool of Pax7+ cells is not replenished by proliferation. Furthermore, a proportion of the Pax7+ cells continuously differentiates and expresses MyoD/MyoG, and therefore the number of Pax7+ cells decreases over time. The level of MyoD/MyoG expression and the proportions of MyoG+ cells are however comparable to control mice, and therefore we do not refer to this as 'premature' differentiation.

Subsequent to the decrease in muscle stem cells, growth of the overall postnatal muscle and fiber diameter was also blunted in coPtpn11 mutant mice, which was accompanied by a major deficit in the accretion of nuclei to fibers. Growth of muscle and fiber diameter also depends on accretion of nuclei, and we therefore suggest that the impairment is, to a large part, due to the observed stem cell deficit (*White et al., 2010*). Mutation of *Ptpn11* in the muscle fiber results in a late-onset atrophy (*Fornaro et al., 2006*; *Princen et al., 2009*). It is possible that atrophy might contribute to the deficit

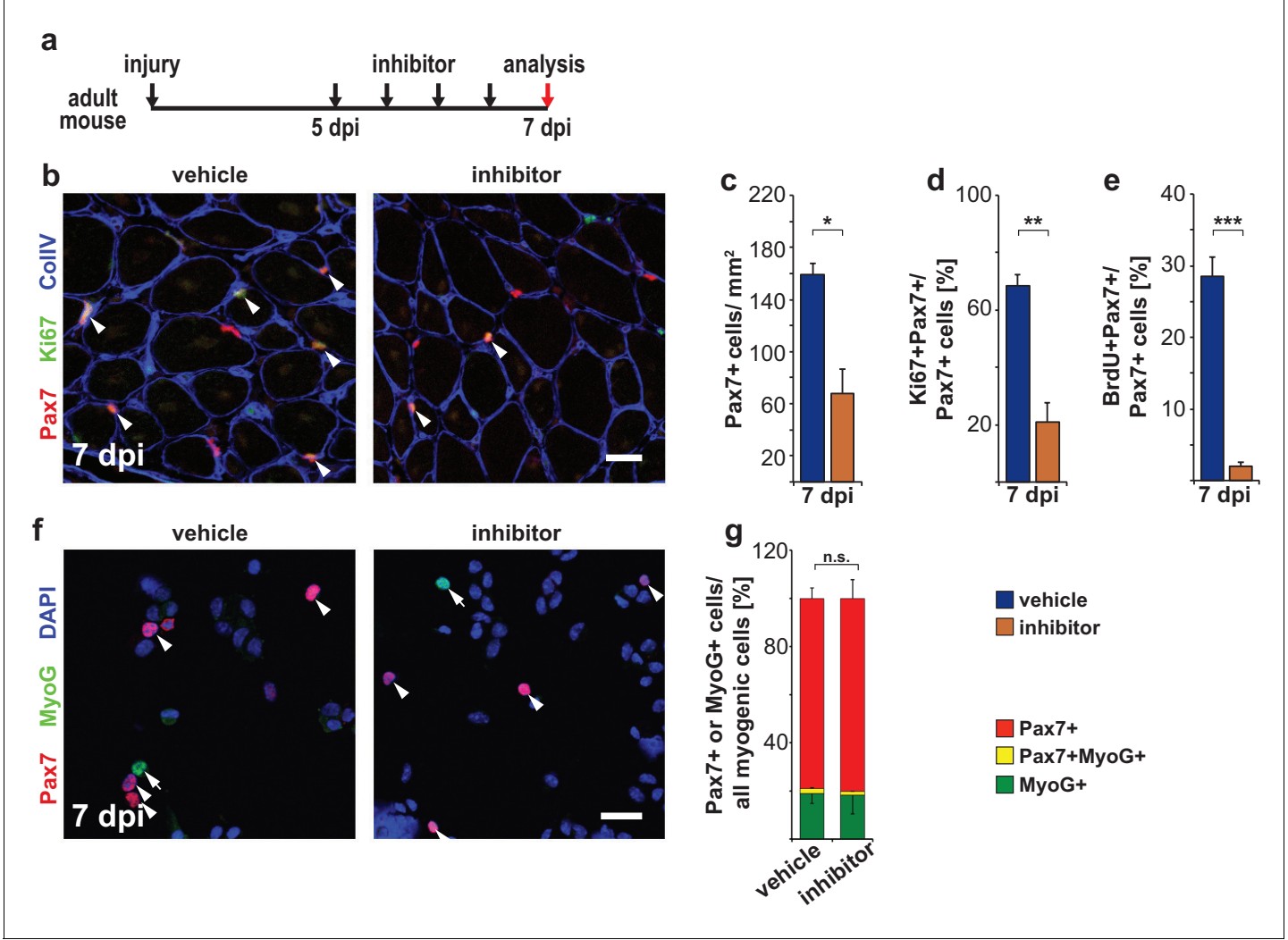

**Figure 8.** Pharmacological inhibition of Ptpn11 leads to a premature cell cycle withdrawal of satellite cells at late stages of muscle regeneration. (**a**) Outline of the experiment. (**b**) Immunostaining for Pax7 (red), Ki67 (green) and collagen IV (CollV; blue) in regenerating *tibialis anterior* muscle seven days after injury (7dpi). Mice were treated with GS493 or with vehicle as indicated. Arrowheads point towards Pax7+Ki67+ cells. (**c–e**) Quantifications of Pax7+ cells/mm$^2$ (**c**), Pax7+Ki67+ cells (**d**) or Pax7+BrdU+ cells (**e**). (**f**) Cytospins of single cells isolated from injured muscle seven days after injury stained for Pax7 (red), MyoG (green) and DAPI (blue); arrowheads and arrows point to Pax7+ and MyoG+ cells, respectively. (**g**) Quantification of Pax7 and MyoG expressing myogenic cells (i.e. Pax7+ plus MyoG+ cells). n.s. = not significant, *p<0.05, **p<0.01, ***p<0.001. Error bars show S.E.M. Scale bars: 25 µm (**b**), 20 µm (**f**).

The following figure supplement is available for figure 8:

**Figure supplement 1.** Pharmacological inhibition of Ptpn11 at late stages of muscle regeneration does not induce apoptosis or adipogenesis.

in muscle growth since most nuclei in the postnatal muscle fibers are expected to lack Ptpn11 in our model.

## Satellite cell activation and muscle repair

Protein translation is repressed in quiescent satellite cells due to phosphorylation of the translation initiation factor eIF2α (*Zismanov et al., 2016*; *Crist et al., 2012*). This repression is relieved when satellite cells get activated, and the appearance of MyoD protein is a hallmark of this. We used myofiber culture to assess the initial stages of satellite cell activation in vitro, and observed that MyoD appears when Ptpn11 is lacking. In contrast, the normal rise in pErk1/2 protein observed in activated

satellite cells was blunted. The appearance of MyoD indicates that Ptpn11 signaling is dispensable for the entry into an early activation stage.

Furthermore, we show that Ki67 expression is initiated in *Ptpn11* mutant satellite cells in myofiber cultures, indicating that satellite cells are able to enter the cell cycle. However, Ki67 protein levels are lowered and expression is not maintained. In accordance, the size of clones observed indicates that only few cells are able to divide, and instead the cells quickly re-enter a resting state. Our genome wide expression analysis indicates that markers for various stages of the cell cycle were downregulated in *Ptpn11* mutant satellite cells ($G_1$/S, S, S/$G_2$, $G_2$/M, M). It is interesting to note that the proliferation deficit can be rescued by a mild activation of the Mapk/Erk1/2 pathway, indicating that it is caused by impaired Mapk/Erk1/2 signaling.

## Ptpn11 and satellite cell quiescence

We demonstrate here that ablation as well as short-term pharmacological inhibition of Ptpn11 in vivo drives Pax7+ cells into a resting state, and this is observed in postnatal development and at late stages of muscle regeneration. Myogenic cells enter quiescence either as Pax7+ progenitor/satellite cells, a process that is reversible, or during terminal differentiation when they express MyoG, which is irreversible. Our data indicate that the cell cycle exit of satellite cells in early postnatal or regenerating muscle after Ptpn11 inhibition/ablation is not accompanied by changes in the expression of differentiation markers in vivo. Thus, Pax7+ cells (re-)entered quiescence prematurely in postnatal development and muscle repair. We did, however, note that *Ptpn11* mutation enhanced differentiation in satellite cells on cultured myofibers. Thus, in regards to the effect on differentiation, differences can be observed in vivo and in vitro. A possible explanation for the diverging differentiation propensity is that satellite cells encounter distinct sets of growth factors in the in vivo environment and in culture.

We show here that Ptpn11 controls entry into quiescence of emerging satellite cells in postnatal development. Our data indicate that downregulation of sustained pErk activity suffices to drive these stem cells out of the cell cycle. In addition to Ptpn11, the well-known target genes of Notch signaling, *Hey/Heyl*, control entry into quiescence, and sex hormones were recently noted to upregulate Notch signaling components and withdrawal of the cell cycle at puberty (*Fukada et al., 2011*; *Kim et al., 2016*). It is possible that Ptpn11/Mapk/Erk1/2 and Notch signaling strengths are modulated in the postnatal muscle and that these signaling systems interact to control entry into quiescence in emerging postnatal satellite cells.

# Material and methods

## Animals

The generation of *Ptpn11^tm1.1Wbm* (called *Ptpn11^flox*), *Pax7^tm1(cre)Mrc* (called *Pax7^Cre*), *Pax7^tm2.1(cre/ERT2)Fan* (called *Pax7^CreERT2*), *Gt(Rosa)26Sor^tm8(Map2k1*,EGFP)Rsky* (called *Map2k1^DD*) and *Gt(ROSA)26Sor^tm1(EYFP)Cos* (called *Rosa^eYFP*) strains were described (*Grossmann et al., 2009*; *Keller et al., 2004*; *Lepper et al., 2009*; *Srinivasan et al., 2009*; *Srinivas et al., 2001*). All experiments were conducted according to regulations established by the Max-Delbrueck-Centre for Molecular Medicine (MDC) and the Landesamt für Gesundheit und Soziales (LAGeSo, Berlin). The strains were maintained on a 129 and C57BL/6 mixed genetic background. For BrdU pulse-chase experiments, animals were sacrificed 2 hr after injection (75 µg/g body weight, Sigma). Ptpn11 inhibitor GS493 (35 µg/g body weight) was used as described (*Grosskopf et al., 2015*). 100 µl of tamoxifen (20 mg/ml; MP Biomedicals, Santa Ana, USA) was injected every 24 hr for five days starting from P35. Cardiotoxin injuries were introduced by intramuscular injection of 40 µl cardiotoxin/PBS solution (10 µM) into the *tibialis anterior* muscle.

## Immunohistology, quantifications of immunohistology and cell counts

Immunohistology was performed as described (*Bröhl et al., 2012*). The following primary antibodies were used: guinea pig-anti-Pax7 (*Bröhl et al., 2012*), mouse-anti-Pax7 (DHSB, Iowa City, USA), rat-anti-BrdU (Biorad, Hercules, USA) rabbit-anti-Ki67 (Leica, Wetzlar, Germany), rabbit-anti-laminin (Sigma-Aldrich, Munich, Germany), goat-anti-collagen (CollV, Millipore, Billerica, USA) goat-anti-desmin, goat-anti-VCAM1 (R and D systems, Minneapolis, USA), rabbit-anti-pErk1/2, rabbit-anti-Erk,

rabbit-anti-pAkt, rabbit-anti-Akt, rabbit-anti-p-p38, rabbit-anti-p38, rabbit-anti-C/EBPα, rabbit-anti-β-actin (Cell Signaling technology, Danvers, USA), rabbit-anti-MyoG, goat-anti-MyoG, rabbit-anti-Ptpn11 (Santa Cruz Biotechnology, Dallas, USA), mouse-anti-Gapdh (Abcam, Cambridge, UK). Cy2/Cy3/Cy5- or peroxidase-conjugated secondary antibodies (Dianova, Hamburg, Germany) were used to detect primary antibodies. For cell counts using Pax7, MyoD and MyoG antibodies, nuclei were always counterstained with DAPI (Sigma-Aldrich, Munich, Germany). In the adult, satellite cells were identified by co-expression of VCAM1 and Pax7, DAPI signal, and location under the basal lamina. Apoptotic cells were detected using the in situ cell death detection kit (Roche, Mannheim, Germany) according to manufacturers instructions.

To quantify pErk levels in floating fiber culture, the area of the satellite cells were outlined using the VCAM1+ signal. The fluorescence intensity of the pErk1/2 staining was quantified in this area using Z-stack images and ImageJ (V1.54S, NIH). The signal per unit area was then determined (integrated fluorescence value/number of pixels determined by VCAM1 staining). This value was corrected for background that was determined using the pErk1/2 signal adjacent to each VCAM1+ cell. More than of 50 cells/animal and cells from at least three animals/genotype were quantified.

## RNA isolation and semi-quantitative PCR

Cells were collected in Trizol (Ambion, Austin, USA) and total RNA was isolated according to manufacturers instructions. cDNA was generated using Superscript III reverse transcriptase (Invitrogen, Carlsbad, USA) according to manufacturers instructions. Semi-quantitative PCR was performed on a Biorad C1000 thermal cycler using the 2x SG qPCR Mastermix (Thermo Fisher Scientific, Waltham, USA) with the following primer pairs:

Ptpn11(exon2,3): For AGTGGAGAGAGGGAAGAGCA Rev AAAGTGGTACTGCCAGACGG;
Ptpn11(exon4,5): For CCGTCTGGCAGTACCACTTT Rev ACAGTCCACACCTTTCTCTCG;
Pax7: For CATGAACCCTGTCAGCAAT Rev CACTGTAGCCAGTGGTGCTG;
MyoD: For GCCCGCGCTCCAACTGCTCTGA Rev CCTACGGTGGTGCGCCCTCTGC;
MyoG: For GGGCCCCTGGAAGAAAAG Rev AGGAGGCGCTGTGGGAGT;
Atg5: For TGTGCTTCGAGATGTGTGGTT Rev GTCAAATAGCTGACTCTTGGCAA;
LC3: For AAAGAGTGGAAGATGTCCGGC Rev GGTCAGGCACCAGGAACTTG;
Ulk1: For AAGTTCGAGTTCTCTCGCAAG Rev CGATGTTTTCGTGCTTTAGTTCC;
Prdm16: For TGCTGACGGATACAGAGGTGT Rev CCACGCAGAACTTCTCGCTAC;
CD31: For GGAAGTGTCCTCCCTTGAGC Rev GAGCCTTCCGTTCTTAGGGT;
Ki67: For ATCATTGACCGCTCCTTTAGGT Rev GCTCGCCTTGATGGTTCCT;
Mcm2: For ATCCACCACCGCTTCAAGAAC Rev TACCACCAAACTCTCACGGTT;
p16: For CCCAACGCCCCGAACT Rev GCAGAAGAGCTGCTACGTGAA;
p21: For TCCACAGCGATATCCAGACA Rev CAGGGCAGAGGAAGTACTGG;
p27: For TTCGACGCCAGACGTAAACA Rev TGCGCAATGCTACATCCAATG;
p57: For GGACCTTTCGTTCATGTAGC Rev ACATGGTACAGAGTGTTCTCA;
p53: For TGAAACGCCGACCTATCCTTA Rev GGCACAAACACGAACCTCAAA
Gapdh: For TGGCAAAGTGGAGATTGTTGCC Rev AAGATGGTGATGGGCTTCCCG

## Whole genome transcriptional profiling and GO term enrichment analysis

RNA from FACS-isolated neonatal VCAM1+CD31-CD45-Sca1- myogenic progenitor cells was used for cRNA synthesis according to manufacturers instructions (Illumina; Illumina total prep, Ambion, Austin, USA). Microarray analysis was done using MouseRef-8 v2.0 Expression BeadChips (Illumina, San Diego, CA). Data were analyzed using GenomeStudio v2010.1 (Illumina, San Diego, CA) and the Partek Genomics Suite (Partek, St. Louis, MO). Deregulated genes were analyzed for enriched GO-terms with the GOrilla database (*Eden et al., 2009*). Microarray data have been deposited in the Gene Expression Omnibus (GEO) database under accession number GSE97430.

## FACS

Isolation of myogenic progenitor cells and adult satellite cells was performed as described (*Bröhl et al., 2012*; *Liu et al., 2013*). Cells were sorted with an Aria II or Aria III FACS sorter (Beckton Dickenson, Franklin Lakes, USA) according to their VCAM1, CD31, CD45 and Sca1 expression or,

alternatively, YFP + cells were isolated from animals carrying the *Rosa^eYFP* reporter allele. Dead cells were excluded by propidium iodide staining (Molecular Probes, Eugene, USA).

## Cell cycle quantification

Paraformaldehyde-fixed, FACS-isolated YFP+ cells were incubated with PBS containing 0.1% Triton-X-100, 3 mg/mL DNase-free RNase (Roche, Mannheim, Germany) and 30 μg/μL propidium iodide (Molecular Probes, Eugene, USA) for 15 min at 37°C. Cells were directly analyzed on a LSR Fortessa analyzer (Beckton Dickenson, Franklin Lakes, USA). Cell cycle distributions were calculated using the cell cycle tool in FloJo X10.0.7r2.

## Flow cytometric detection of apoptotic satellite cells

Satellite cells were stained with antibodies for VCAM1, CD31, CD45 and Sca1. Apoptotic satellite cells were identified with the Vybrant FAM Casp3 and Casp7 assay Kit (Molecular Probes, Eugene, USA) according to the manufacturers instructions.

## Cell and myofiber culture

C2C12 cells (ATCC, Manassas, USA) were cultured in DMEM supplemented with 20% FCS. Primary myogenic progenitor cells were seeded at a density of 40.000 cells per well and cultured for one day in DMEM/F12 (1:1) supplemented with 15% FCS on 16-well Labtek chamberslides (Nunc, Roskilde, Denmark) coated with 10% Matrigel (Sigma-Aldrich, Munich, Germany). For the in vitro differentiation assay, cells were seeded at a concentration of 120.000 cells per well, cultured overnight in DMEM/F12 (1:1) supplemented with 15% FCS and switched to differentiation medium (DMEM supplemented with 2% horse serum). Floating single myofiber cultures were performed as described (*Vogler et al., 2016*). Cultures were incubated with 10 μM BrdU for one hour. Quantifications of nuclei in fibers of the *extensor carpi radialis longus* muscle were performed as described (*Bröhl et al., 2012*).

## Statistics

Three or more animals were used per genotype and experiment. Data were analyzed using an unpaired, 2-tailed T-test. p-values below 0.05 were considered significant. Results are shown as arithmetical mean ± standard error of the mean (S.E.M.). n.s.: not significant; p>0.05; *p<0.05; **p<0.01; ***p<0.001

## Acknowledgement

We thank Petra Stallerow and Claudia Päseler for animal husbandry, Vivian Schulz, Pia Blessin and Bettina Brandt for technial support, Gabriele Born for Illumina array hybridization, Hans Peter Rahn for advice and help with FACS sorting, Walter Birchmeier and Thomas Müller (all at the MDC, Berlin) for critical reading of the manuscript. This work was supported by a fellowship to JG (MyoGrad International Graduate School for Myology, GK 1631, Deutsche Forschungsgemeinschaft) and by a grant to CB (Klinische Forschergruppe KFO 192, Deutsche Forschungsgemeinschaft).

## Additional information

### Funding

| Funder | Grant reference number | Author |
| --- | --- | --- |
| Deutsche Forschungsgemeinschaft | Graduate student fellowship | Joscha Griger |
| Deutsche Forschungsgemeinschaft | KFO192 | Simone Spuler<br>Carmen Birchmeier |

The funders had no role in study design, data collection and interpretation, or the decision to submit the work for publication.

## Author contributions

JG, Formal analysis, Investigation, Writing—original draft, Project administration, Writing—review and editing; RS, IL, Formal analysis, Investigation; VS, Investigation; CK, Resources; SS, Supervision; MN, synthesized and provided GS493; CB, Conceptualization, Supervision, Funding acquisition, Writing—original draft, Project administration, Writing—review and editing

## Author ORCIDs

Joscha Griger, http://orcid.org/0000-0001-8666-3371
Simone Spuler, http://orcid.org/0000-0002-0155-1117
Carmen Birchmeier, http://orcid.org/0000-0002-2041-8872

## Ethics

Animal experimentation: All procedures involving animals and their care were carried out in accordance with the guidelines for animal experiments at the Max-Delbrueck-Center (MDC), which conform to the Guide for the Care and Use of Laboratory Animals (NIH Publication No. 85-23, revised 1996), the European Parliament Directive 2010/63/EU and the 22 September 2010 Council on the protection of animals. Animal experimentation was approved by the local Ethics committee for animal experiments at the Landesamt für Gesundheit und Soziales (GO130/13; G0028/14). The animal house at the MDC is registered according to paragraph11 German Animal Welfare Law.

# Additional files

## Supplementary files

• Supplementary file 1. List of all deregulated genes in neonatal Ptpn11 mutant muscle stem cells. Transcripts labeled in colors are also shown in *Table 1* and are associated with particular GO terms. Green: cell cycle-associated; yellow: immediate early gene; purple: extracellular matrix and matrix remodeling. FDR: false discovery rate. Corrected p-values (q-value) was determined as described (*Hochberg and Benjamini, 1990*).

• Supplementary file 2. List of enriched GO terms. Enriched GO terms in deregulated genes in Ptpn11 mutant muscle stem cells. FDR: false discovery rate. Corrected p-values (q-value) was determined as described (*Hochberg and Benjamini, 1990*).

• Supplementary file 3. Statistical data for each experiment shown in *Figures 1–8* and *Figure 1—figure supplement 1*; *Figure 2—figure supplement 1*; *Figure 2—figure supplement 2*; *Figure 4—figure supplement 1*; *Figure 5—figure supplement 1*; *Figure 6—figure supplement 1*; *Figure 7—figure supplement 1*; *Figure 8—figure supplement 1*. Shown are the definition of the center, the error bars, the statistical test used, n-numbers for each genotype or treatment for a specific developmental stage and the corresponding p-value for each comparison. S.E.M.: standard error of the mean.

## Major datasets

The following dataset was generated:

| Author(s) | Year | Dataset title | Dataset URL | Database, license, and accessibility information |
|---|---|---|---|---|
| Griger J | 2017 | Genome wide expression analysis of FACS-isolated Shp2 mutant neonatal myogenic progenitor cells | https://www.ncbi.nlm.nih.gov/geo/query/acc.cgi?acc=GSE97430 | Publicly available at the NCBI Gene Expression Omnibus (accession no: GSE97430) |

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
