## [Decision Letter]

Thank you for submitting your article "Loss of Shp2 (Ptpn11) drives satellite cells into quiescence" for consideration by *eLife*. Your article has been reviewed by two peer reviewers, and the evaluation has been overseen by a Reviewing Editor and Fiona Watt as the Senior Editor. The following individual involved in review of your submission has agreed to reveal his identity: So-ichiro Fukada (Reviewer #1).

The reviewers have discussed the reviews with one another and the Reviewing Editor has drafted this decision to help you prepare a revised submission.

This paper shows the essential role of Shp2, a non-receptor tyrosine phosphatase, in muscle stem cells during postnatal development and adult muscle regeneration The authors also show that Shp2 is required for the sustained activation of Mapk/Erk1/2 in myogenic cells. Their results are complemented by additional mouse models and pharmacological approaches to modulate MAPK (ERK) signalling. These are novel and interesting results, however a number of points need to be addressed by the authors.

Most importantly the authors need to clarify why quiescent satellite cells are reduced in the mutant and why there is a proliferation defect. As the experiments are done, it may well be that satellite cells do not even get activated efficiently. In addition, with the data presented, an apoptosis phenotype cannot be ruled out.

More detailed comments are as follows:

1) Apoptosis:

In Figure 2 reduction in the number of satellite cells lacking Shp2 is shown at P0 and P7. In Figure 2—figure supplement 1, the same number of Tunel+ cells per mm2 is quantified. Wouldn't this mean that there is a higher percentage of apoptotic cells in the KO mice? In addition, in this panel: Tunel+ cells do not necessarily indicate satellite cells. Are these cells Pax7+? Even if the mouse is a conditional mutant for satellite cells, other cell types could be altered (apoptotic) by the reduction of satellite cell number in conditions of muscle injury.

Similarly, a comparison of the reduction of satellite cells in Figure 5–Figure 6 and the quantification of Tunel+ satellite cells in Figure 6—figure supplement 1 needs to be recalculated, and the influence (or not) of apoptosis re-evaluated.

2) Activation – Proliferation:

In Figure 4, satellite cells lacking Shp2 do not form colonies (Figure 4). The authors indicate that they do not incorporate BrdU (Figure 4). There are also less Ki67+ satellite cells (Figure 4). This leads to the conclusion that Shp2-null satellite cells have a proliferation defect.

Two comments to this issue:

If the satellite cells do not proliferate (they do not form colonies, or colonies only have 1 cell -see Figure 4), then it is not really clear that the cells do not incorporate BrdU. Actually, the few satellite cells remaining (Figure 4) could all be incorporating BrdU. Can the authors improve these data?

In addition, the staining of Ki67 (Figure 4) is not nuclear. Thus, are the satellite cells entering the cell cycle at all in the absence of Shp2? It may well be that they cannot reach the division point. Therefore, the conclusion that the satellite cells null for Shp2 cannot proliferate well, and they return to quiescence is a bit tight. The cells might not ever leave the quiescent state, or get activated. This set of experiments needs to be better sustained experimentally to support the conclusions, as they seem a bit contradictory.

3) Figure 1: Shp2 plays a role in muscle hypertrophy. Therefore, is there any possibility that the loss of Shp2 in myofibers directly influences their size? Since the authors indicated the decreased myonuclei number in coShp2 mice, the loss of the satellite cell pool appears to have an impact on the decreased size of myofibers. If Shp2 has a direct effect on myofiber, the authors should discuss the dependency of Shp2 in embryonic or postnatal myofibers. As well as satellite cells, the dependency of Shp2 might be different between embryonic and postnatal myofibers.

4) Figure 5: Can the authors postulate on the origin of adipogenesis in the absence of Shp2 in satellite cells? What is the phenotype in the MEK1DD mutant mice?

If the defect of the satellite cell is activation- or cell-cycle-related, the time-points of analysis after muscle injury of 7 and 10 days are not adequate. The 7-10 time point are valid for analysis of regenerating muscle fibres, not for the prior satellite cell activation/proliferative status. 1 to 3-4 days post injury should be analysed to support the satellite cell conclusions.

Figure 5; Data clearly show the decreased number of muscle stem cells. There are three ways to explain the loss of the muscle stem cell pool; a) Cell death including apoptosis, b) Premature differentiation, c) Conversion of muscle stem cells to non-myogenic cells. However, those mechanisms do not apply to this case. The authors should discuss how the number of muscle stem cells is regulated when Shp2 is conditionally abrogated in adult muscle stem cells.

In addition, the result of decreased number of satellite cells in TxcoShp2 is interesting. Why was the number of more quiescent satellite cells reduced? Did most of the satellite cells divide at least one time during 90 days? In this case, do the remaining satellite cells in TxcoShp2 have any relevance with the LRC (label retaining cells) described in Chakkalakal et al. Nature 2012? Furthermore, is the slow-dividing event essential for maintenance of the muscle satellite cell pool?

5) Figure 6: After GS493 treatment: is there an adipogenic phenotype in the muscle?

6) Figure 3: If satellite cells lacking Shp2 cannot proliferate (the cells show blunted clonogenic potential in Figure 4), how did the authors reach to get so many myoblasts to perform the experiments in Figure 3 – differentiation assays? Are these myogenic precursor cells subjected to passaging in vitro? Are they freshly isolated? Could the authors explain the experimental approach followed?

Figure 3: Can the authors test whether satellite cells lacking Shp2 are indeed negative for Ki67 and that they do not proliferate?

---

## [Author Response]

*[…] Most importantly the authors need to clarify why quiescent satellite cells are reduced in the mutant and why there is a proliferation defect. As the experiments are done, it may well be that satellite cells do not even get activated efficiently. In addition, with the data presented, an apoptosis phenotype cannot be ruled out.*

We observe a loss of satellite cells in the adult resting muscle that occurs exceedingly slow, as well as a fast one during injury/regeneration and in the early postnatal period. We re-investigated carefully different possible mechanisms, apoptosis, autophagy and (trans-)differentiation. We discuss each mechanism separately to address the major comment (mechanisms in the adult phase); we include in this discussion also specific comments of the reviewers that relate to this.

Sort summary and conclusion of our experiments:

The major phenotype that we observe is the withdrawal from the cell cycle/impaired proliferation. Thus, the pool of Pax7+ cells is not replenished by proliferation. Furthermore, a proportion of the Pax7+ cells continuously differentiates and expresses MyoD/MyoG, and therefore the number of Pax7+ cells decreases over time. The level of MyoD /MyoG expression and the proportions of MyoG+ cells are however comparable in control mice, and therefore we do not refer to this as ‘premature’ differentiation. It is possible that in the adult uninjured muscle a similar mechanism is at work. However, the loss of cells is slow (0.04 satellite cells/100 fibers/day) and small changes might thus account for it. Possibly, the low cell cycle activity is needed for maintenance of the satellite cell pool, but because of small numbers we cannot exclude other mechanisms. This is now commented on in the revised manuscript (subsection “Ptpn11 controls proliferation of adult satellite cells and muscle repair”, third paragraph).

In the acutely injured muscle of TxcoShp2 mice, we see in addition to the proliferation deficit an increased apoptosis at early time points after injury. This is the only condition where we observed apoptosis. These experiments were suggested by the reviewer, are included in Figure 7 of the revised manuscript, and are now mentioned in Abstract, Materials and methods, and Discussion.

*More detailed comments are as follows:*

*1) Apoptosis:*

*In Figure 2 reduction in the number of satellite cells lacking Shp2 is shown at P0 and P7. In Figure 2—figure supplement 1, the same of Tunel+ cells per mm2 is quantified. Wouldn't this mean that there is a higher percentage of apoptotic cells in the KO mice? In addition, in this panel: Tunel+ cells do not necessarily indicate satellite cells. Are these cells Pax7+? Even if the mouse is a conditional mutant for satellite cells, other cell types could be altered (apoptotic) by the reduction of satellite cell number in conditions of muscle injury.*

1a) Apoptosis in the early postnatal period.

Quantification of TUNEL+ cells is shown in the revised manuscript as (i) percentage of Pax7+ cells that are TUNEL+; (ii) TUNEL+ cells/mm2 (revised manuscript Figure 2—figure supplement 1). In addition, TUNEL staining was performed in the postnatal muscle after 2 day treatment with the GS493 inhibitor and quantified as TUNEL+ Pax7+/Pax7+ cells and as TUNEL+ cells/mm2 (revised manuscript Figure 2—figure supplement 1). We observed in none of these assays a substantial increase in apoptotic cells. Compared to the adult situation, the loss of satellite cells in the postnatal period is fast (i.e. estimated loss of around 4 cells/100 myofibers/day between E16 and P7). We are therefore confident that we can exclude apoptosis as the mechanism responsible for the disappearance of Pax7+ cells in the early postnatal period.

b) Apoptosis in the adult resting muscle.

We observe a slow loss of satellite cells in the resting muscle of the adult after Shp2 mutation, i.e. a loss of about 70% of the cells over a 90 day period (loss of 0.04 satellite cells/100 fibers/day). Apoptosis appears not to account for this, since neither TUNEL staining nor a caspase activity assay done by FACS revealed changes in apoptosis (revised manuscript Figure 6—figure supplement 1).

c) Apoptosis during the acute/inflammatory phase of injury.

As suggested by the reviewer, we analyzed apoptosis during early time points after injury, which revealed 2.5 days after injury an increased number of apoptotic cells (revised manuscript Figure 7).

d) Apoptosis during the late phase or regeneration.

*Similarly, a comparison of the reduction of satellite cells in Figure 5–Figure 6 and the quantification of Tunel+ satellite cells in Figure 6—figure supplement 1 needs to be recalculated, and the influence (or not) of apoptosis re-evaluated.*

To assess the function of Shp2 in the late phase of regeneration, we used short-term treatment using the GS493 inhibitor. Quantification of TUNEL+ cells is shown in the revised manuscript as (i) percentage of Pax7+ cells that are TUNEL+; (ii) TUNEL+ cells/mm2 (revised manuscript Figure 8—figure supplement 1).

Autophagy/senescence

We observe no evidence for autophagy as assessed by analysis of transcriptionally regulated genes in autophagy (revised manuscript Figure 2—figure supplement 2; Figure 6). Furthermore, p16 and p53 up-regulation indicative of senescence was not observed (revised manuscript; Figure 6).

Differentiation/Transdifferentiation

A change in the propensity of myogenic differentiation would be accompanied by upregulated MyoD or MyoG (either protein or transcript level). We do not observe this in vivo. Transcripts of MyoD/MyoG were analyzed in coShp2 mutant satellite cells at P0 (revised manuscript Figure 2—figure supplement 2), MyoG protein and differentiation capacity at P0 (revised manuscript Figure 3), proportion of MyoG+ cells after GS493 treatment at P8 (revised manuscript Figure 3), MyoD/MyoG transcripts in satellite cells of adult resting muscle (revised manuscript Figure 6), or proportion of MyoG+ cells in late injury after GS493 treatment (Figure 8). These analyses have all been done multiple times and there is no doubt about their outcome. We therefore exclude ‘premature’ differentiation.

We detect no evidence for trans-differentiation in the postnatal or adult satellite cells at P0 (revised manuscript Figure 2—figure supplement 2; revised manuscript Figure Dd; revised manuscript Figure 7—figure supplement 1). However, in regeneration experiments, we observe adipogenesis in the muscle (revised manuscript Figure 7 and Figure 7—figure supplement 1). Previous observations suggest that adipogenesis is a general outcome when muscle cannot be regenerated e.g. in mouse models that lack satellite cells or delay formation of new myofibers, (see for instance Sambasivan, Development. 2011 138(17):3647-56; Lepper, Development. 2011 138(17):3639-46; Uezumi, Nat Cell Biol. 2010 12(2):143-52.). This was previously assigned to differentiation of FAPs. We therefore propose that adipogenesis is caused by indirect mechanisms which is now mentioned on in the revised manuscript (subsection “Ptpn11 controls proliferation of adult satellite cells and muscle repair”, third paragraph.

*2) Activation – Proliferation:*

*In Figure 4, satellite cells lacking Shp2 do not form colonies (Figure 4). The authors indicate that they do not incorporate BrdU (Figure 4). There are also less Ki67+ satellite cells (Figure 4). This leads to the conclusion that Shp2-null satellite cells have a proliferation defect.*

Signs of activation of normal satellite cells in floating fiber culture are the appearance of MyoD, pErk1/2 and Ki67. In Shp2 mutant cells we observe in >90% of the satellite cells MyoD appearance, and a blunted pErk1/2 upregulation. Furthermore, Ki67 and incorporation of BrdU are observed in a subpopulation. The numbers indicate that this subpopulation (around 50%) undergoes one round of division during 3 days of incubation (Figure 5 and Figure 5—figure supplement 1).

*Two comments to this issue:*

*If the satellite cells do not proliferate (they do not form colonies, or colonies only have 1 cell -see Figure 4), then it is not really clear that the cells do not incorporate BrdU. Actually, the few satellite cells remaining (Figure 4) could all be incorporating BrdU. Can the authors improve these data?*

The colony size after 72 hours is 1.5- i.e. half of the cells have divided once, and in accordance about half of the cells have initiated Ki67 (revised manuscript Figure 5).

*In addition, the staining of Ki67 (Figure 4) is not nuclear. Thus, are the satellite cells entering the cell cycle at all in the absence of Shp2? It may well be that they cannot reach the division point.*

The picture displaying Ki67 staining was exchanged. Our data indicate that a subpopulation of Shp2 mutant cells expresses nuclear Ki67 at 48hrs, but levels are lower than in control cells (revised manuscript Figure 5, quantified in B). At 72hrs the fraction of Ki67+ cells is even smaller. Array analyses and PCR done on the postnatal satellite cells indicated that markers of various stages of the cell cycle are downregulated (G1/S-specific cyclin-E2; G1/S, S and M-specific Ranbp1; S, G2 and M phase specific Prc1; G2/M-phase specific cdk1; M-phase-specific cyclin B1 and Nusap1). This is now mentioned in the last paragraph of the subsection “Ptpn11 controls myogenic stem cell proliferation in postnatal mice” and in the subsection “Ptpn11 and satellite cell quiescence” of the revised manuscript, and supported by a cell cycle analysis (revised manuscript Figure 2—figure supplement 2).

*Therefore, the conclusion that the satellite cells null for Shp2 cannot proliferate well, and they return to quiescence is a bit tight. The cells might not ever leave the quiescent state, or get activated.*

Since >90% of the satellite cells turn on MyoD in floating fiber cultures, they are entering an early stage of activation. A subpopulation initiates Ki67 expression, but they proliferate inefficiently and quickly downregulate cell cycle markers (Ki67). We independently analyze satellite activation after injury in vivo using Ki67 and pHH3 as markers for cell cycle entry and M-phase, respectively. This demonstrated that most satellite cells 2.5 days after muscle injury enter the cell cycle (Ki67 in around 70% of the cells), but few enter M-phase (pHH3 is low; revised manuscript Figure 7). Thus, satellite cells are activated in vivo but they do not proliferate efficiently. We conclude from these series of experiments that satellite cells are transiently activated in vivo and in vitro.

*This set of experiments needs to be better sustained experimentally to support the conclusions, as they seem a bit contradictory.*

Signs of activation of normal satellite cells in floating fiber culture are the appearance of MyoD, pErk1/2 and Ki67. In Shp2 mutant cells we observe in >90% of the satellite cells MyoD appearance, and a blunted pErk1/2 upregulation. Furthermore, Ki67 and incorporation of BrdU are observed in a subpopulation. The numbers indicate that this subpopulation (around 50%) undergoes one round of division during 3 days of incubation (Figure 5 and Figure 5—figure supplement 1).

The colony size after 72 hours is 1.5- i.e. half of the cells have divided once, and in accordance about half of the cells have initiated Ki67 (revised manuscript Figure 5).

*3) Figure 1: Shp2 plays a role in muscle hypertrophy. Therefore, is there any possibility that the loss of Shp2 in myofibers directly influences their size? Since the authors indicated the decreased myonuclei number in coShp2 mice, the loss of the satellite cell pool appears to have an impact on the decreased size of myofibers. If Shp2 has a direct effect on myofiber, the authors should discuss the dependency of Shp2 in embryonic or postnatal myofibers. As well as satellite cells, the dependency of Shp2 might be different between embryonic and postnatal myofibers.*

Others analyzed Shp2 function in the adult muscle, and observed a late onset atrophy after loss of Shp2 (11 week old mice: Fornaro, J Cell Biol. 2006 175(1):87-97). We observe a change in fiber diameter after stem cells and myonuclei numbers are reduced. We thus conclude that the atrophy is due to a deficit in the accretion of nuclei (Figure 1).

The reviewer raises the possibility that the loss of Shp2 has distinct consequences on postnatal and embryonic fibers. The cre line used (Pax7^Cre^) starts to recombine around E12 in muscle progenitors. Syncytial fibers contain thus probably a (small) proportion of unrecombined nuclei. Therefore, a strong statement on a stage-specific role of Shp2 in atrophy is not possible.

*4) Figure 5: Can the authors postulate on the origin of adipogenesis in the absence of Shp2 in satellite cells?*

Adipogenesis is typically observed when muscle regeneration fails. Thus, we assign the pronounced adipogenesis as secondary to the failed muscle regeneration, possibly to a differentiation of FAPs as described by others (see for instance Sambasivan, Development. 2011 138(17):3647-56; Lepper, Development. 2011 138(17):3639-46; Uezumi, Nat Cell Biol. 2010 12(2):143-52.). This is mentioned in the third paragraph of the subsection “Ptpn11 controls proliferation of adult satellite cells and muscle repair”.

*What is the phenotype in the MEK1DD mutant mice?* Our preliminary data indicate that 10 days after tamoxifen (i.e. induction of MEK1DD expression) satellite cell numbers were unchanged and not activated (no significant difference in the proportion of Ki67+ cells as in control animals). We tested regeneration in two animals (7 days after regeneration), and observed increased Pax7+ cells/mm^2^. In addition, similar to the in vitro situations these cells do not efficiently differentiate in vivo.

*If the defect of the satellite cell is activation- or cell-cycle-related, the time-points of analysis after muscle injury of 7 and 10 days are not adequate. The 7-10 time point are valid for analysis of regenerating muscle fibres, not for the prior satellite cell activation/proliferative status. 1 to 3-4 days post injury should be analysed to support the satellite cell conclusions.*

As suggested by the reviewer, we analyzed early stages of regeneration. This revealed an activation of satellite cells (appearance of Ki67 expression), a deficit in M-phase entry (strongly reduced pHH3 compared to controls), and a loss of satellite cells due to apoptosis (Figure 7). After short-term pharmacological inhibition at late stages of regeneration, premature withdrawal of the cell cycle was the only observable phenotype (revised manuscript Figure 8; Figure 8—figure supplement 1).

*Figure 5; Data clearly show the decreased number of muscle stem cells. There are three ways to explain the loss of the muscle stem cell pool; a) Cell death including apoptosis, b) Premature differentiation, c) Conversion of muscle stem cells to non-myogenic cells. However, those mechanisms do not apply to this case. The authors should discuss how the number of muscle stem cells is regulated when Shp2 is conditionally abrogated in adult muscle stem cells.*

The rate of satellite cells loss is small in the non-injured muscle when we compare Pax7cre and Pax7creShp2flox/flox mice after tamoxifen: about 70% of the cells disappear have disappeared after a 3-month period, i.e. 0.04 satellite cells/100 fibers/day. We find no evidence for cell death (TUNEL staining, caspase activity assay using FACS), no evidence for autophagy or senescence (upregulated expression of autophagy- or senescence-related genes, qPCR), no evidence for ‘premature differentiation’, i.e. upregulation of MyoD or MyoG (qPCR). Relatively small amounts of MyoD/MyoG transcripts are present in control and mutants. We hypothesize that this reflects a slow turnover/differentiation i.e. the fact that satellite cells contribute to fibers even in adult sedentary muscle (Pawlikowski, 2015 Skelet Muscle, 5:42; Keefe, 2015 Nat Commun. 6:7087). We do not find evidence for trans-differentiation into adipocytes or endothelia (qPCR). The data raise the possibility that low cell cycle activity plays a role in satellite cell maintenance. However, the very small rate of cell disappearance (i.e. loss of 0.04 satellite cells/100 fibers/day) makes it difficult to exclude alternative mechanisms. The data addressing this are shown in the revised manuscript in Figure 6, and commented on in the revised manuscript (subsection “Ptpn11 controls proliferation of adult satellite cells and muscle repair”, first paragraph).

*In addition, the result of decreased number of satellite cells in TxcoShp2 is interesting. Why was the number of more quiescent satellite cells reduced? Did most of the satellite cells divide at least one time during 90 days? In this case, do the remaining satellite cells in TxcoShp2 have any relevance with the LRC (label retaining cells) described in Chakkalakal et al. Nature 2012? Furthermore, is the slow-dividing event essential for maintenance of the muscle satellite cell pool?*

We attempted to define whether LRC or nonLRC cells preferentially disappear in adult TxcoShp2 mice, and established for this the assay described by Chakkalakal and colleagues. In our hands, the proportions of LRC and nonLRC cells are variable between animals, possibly due the fact that we do not keep the animals on a clean C57/Bl6 background.

*5) Figure 6: After GS493 treatment: is there an adipogenic phenotype in the muscle?*

After GS493 treatment, adipocytes are not increased (revised manuscript Figure 7—figure supplement 1).

*6) Figure 3: If satellite cells lacking Shp2 cannot proliferate (the cells show blunted clonogenic potential in Figure 4), how did the authors reach to get so many myoblasts to perform the experiments in Figure 3 – differentiation assays? Are these myogenic precursor cells subjected to passaging* in vitro*? Are they freshly isolated? Could the authors explain the experimental approach followed?*

The satellite cells were freshly isolated from early postnatal control and coShp2 mice, i.e. at a developmental stage when satellite cells are more abundant than in the adult. In addition, cells were plated at high density and low serum (5% horse serum) to ensure fast differentiation (revised manuscript Figure 3). In contrast, the proliferation assay was performed at 3-fold lower density and in proliferation medium (15% fetal calf serum). This is now described in detail in Materials and methods (subsection “Cell and myofiber culture”).

Figure 3: Can the authors test whether satellite cells lacking Shp2 are indeed negative for Ki67 and that they do not proliferate?

We tested this in postnatal and adult muscle using Ki67 immunohistology, qPCR, Ki67 and MCM2 transcripts/protein, analysis of other cell cycle markers and BrdU incorporation, in vitro, in vivo using mutants and pharmacological inhibition (revised manuscript Figure 2; Figure 2—figure supplement 2; Figure 5; Figure 6; Figure 7; Figure 8). The data are unambiguous- satellite cells lacking Shp2 in early postnatal or adult uninjured muscle are mostly negative for Ki67; when activated they express Ki67 for a short period and do not efficiently proliferate.